# PEML: Parameter-efficient Multi-Task Learning with Optimized Continuous Prompts

## Abstract

Parameter-Efficient Fine-Tuning (PEFT) is widely used for adapting Large Language Models (LLMs) for various tasks. Recently, there has been an increasing demand for fine-tuning a single LLM for multiple tasks because it requires overall less data for fine-tuning thanks to the common features shared among tasks. More importantly, LLMs are resource demanding and deploying a single model for multiple tasks facilitates resource consolidation and consumes significantly less resources compared to deploying individual large model for each task. Existing PEFT methods like LoRA and Prefix Tuning are designed to adapt LLMs to a specific task. LoRA and its variation focus on aligning the model itself for tasks, overlooking the importance of prompt tuning in multi-task learning while Prefix Tuning only adopts a simple architecture to optimize prompts, which limits the adaption capabilities for multi-task. To enable efficient fine-tuning for multi-task learning, it is important to co-optimize prompt optimization and model adaptation. In this work, we propose a Parameter-Efficient Multi-task Learning (PEML), which employs a neural architecture engineering method for optimizing the continuous prompts while also performing low-rank adaption for model weights. We prototype PEML by creating an automated framework for optimizing the continuous prompts and adapting model weights. We evaluate PEML against state-of-the-arts multi-task learning methods MTL-LoRA, MultiLoRa, C-Poly, and MoE, on the GLUE, Super-GLUE, Massive Multitask Language Understanding, and commonsense reasoning benchmarks. The evaluation results present an average accuracy improvement of up to 6.67%, with individual tasks showing peak gains of up to 10.75%.

## 1 Introduction

Large language models (LLMs) have made significant advancements in various natural language processing tasks such as machine translation Lewis et al. (2019), text generation Chung et al. (2022), and code analysis Wang et al. (2021); Qin et al. (2024). Traditional task-specific fine-tuning (FT) becomes increasingly computationally expensive as LLMs continue to grow in size. It requires adjusting all of the model's parameters, which makes it difficult to scale Devlin et al. (2018); Howard & Ruder (2018); Raffel et al. (2020), and thus motivated parameter-efficient fine-tuning (PEFT) methods that only require learning a small set of additional parameters for each task Houlsby et al. (2019); Lester et al. (2021). These methods Pfeiffer et al. (2020); Hu et al. (2021) have been widely adopted as they offer comparable performance to full fine-tuning while significantly reducing computational overhead Houlsby et al. (2019); Lester et al. (2021); Ding et al. (2023).

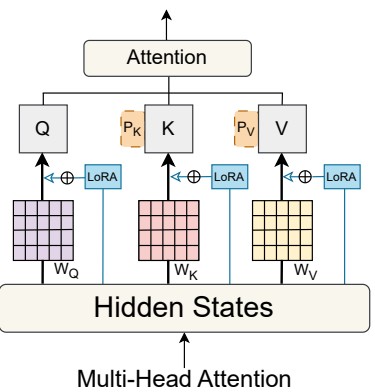

Figure 1: Overview of PEML.

LoRA Hu et al. (2021) and Prefix Tuning Li & Liang (2021) are among prominent PEFT methods for adapting models to a single task. LoRA enhances efficiency by introducing trainable low-rank matrices into a subset of the model's weights during training, enabling learning directional updates in the parameter space. Prefix Tuning improves adaptability by generating task-specific continuous vectors Liu et al. (2021a); Li & Liang (2021) to the input embeddings before each transformer layer to steer the model's generation process without modifying the original model parameters. Such learned prefix vectors and low-rank matrices enable efficient adaptation to new and related tasks with minimal additional training Lester et al. (2021); Liu et al. (2022; 2021b).

LoRA and Prefix Tuning, however, face challenges when applied to multi-task training. First, deploying many task-specific adapters (e.g., prefix vectors or LoRA matrices) increases memory usage and makes resource management complex. Frequent switching between adapters incurs computational costs due to the need for adapter loading and model reconfiguration. Therefore, it is inefficient and costly for inference serving deployment. In addition, individual task training prevents knowledge sharing across tasks, missing opportunities to leverage insights from one task to improve others Lopes et al. (2023); Zamir et al. (2020). Such isolated task training limits potential performance gains from inter-task knowledge-sharing.

Lately, there are efforts to adapt PEFT methods for multi-task learning. MPT Wang et al. (2023d) learns a shared transferable prompt distilled from multiple task-specific prompts and applies multiplicative low-rank adaptations for downstream task specialization. However, it requires pre-training individual teacher prompts for each task. MultiLoRA Wang et al. (2023c) extends LoRA by horizontally scaling modules, dividing them into parallel sub-modules with separate scaling factors. Yet, this approach increases VRAM usage due to activation caching for multiple parallel modules. C-Poly Wang et al. (2023a) employs a skill-based framework that merges shared and task-specific low-rank parameters using a learned skill matrix, but its fixed architecture limits generalization to unseen tasks. MTL-LoRA Yang et al. (2025) introduces task-adaptive parameters that reduce interference in shared low-dimensional spaces. However, it requires task-specific routing during inference, which complicates inference deployment and resource management. Despite these advancements, most approaches focus on extending LoRA but overlook the critical aspect of prompt alignment in multi-task environment Shen et al. (2024); Xin et al. (2024). Aligned prompts can significantly improve model generalization during multi-task Xu et al. (2022) training. Motivated by this observation, we explore integrating prompt alignment into PEFT methods to enhance multi-task performance.

To this end, we propose PrefixNAS which generates and optimizes a single, unified continuous prompt architecture through neural architecture search (NAS) for better alignment of the model's behavior in multi-task learning. PrefixNAS captures task-relevant features and relationships, allowing the prompt encoder to leverage shared knowledge efficiently while preserving task-specific distinctions (∼ see Appendix 7.2.1). Additionally, PrefixNAS automatically tunes both the prefix architecture and its hyperparameters, eliminating the need for manual adjustments when adapting to new tasks. We further develop a Parameter-efficient Multi-Task Learning framework (PEML) to integrate PrefixNAS enabled prompt optimization into LoRA for model alignment. Figure 1 shows the structure of PEML. LoRA matrices are applied to all projection layers, while prefix vectors are added in parallel to only the key and value projections of each attention head. This combination allows the model to adapt to new tasks, with LoRA handling model adaptation and prefix vectors handling input alignment. Such an integrated design also facilitates efficient inference deployment as only one adapter needs to be deployed and does not require adapter switching.

We formulate multi-task learning as a joint optimization problem through LoRA and PrefixNAS and conduct theoretical analysis on PEML. We evaluate PEML by comparing it with state-of-the-arts multi-task learning methods such as MTL-LoRA, MultiLoRa, C-Poly, and MoE Yang et al. (2025); Shazeer et al. (2017); Wang et al. (2023c;a), on the GLUEWang et al. (2018), SuperGLUEWang et al. (2019), Massive Multitask Language Understanding Hendrycks et al. (2021) and commonsense reasoning benchmarks Bisk et al. (2020); Sakaguchi et al. (2020); Mihaylov et al. (2018); Zellers et al. (2019); Clark et al. (2018). The evaluation results demonstrate an average accuracy improvement of up to 6.67%, with individual tasks showing peak gains of up to 10.75%.

## 2 RELATED WORK

Approaches tackling parameter efficient fine tuning can be broadly classified into three categories: adapter-based, prompt-based (e.g. Prefix Tuning), and low-rank adaptation techniques.

**Adapter-based methods** Houlsby et al. (2019); He et al. (2021); Mahabadi et al. (2021) insert small, trainable modules into a pretrained model while keeping the rest of the model frozen, capturing task-specific information with minimal added parameters. It introduce additional layers, leading to parameter redundancy, whereas LoRA focuses on low-rank updates without introducing additional layers

**Prompt-based methods** Lester et al. (2021); Razdaibiedina et al. (2023); Wang et al. (2023b) adjust only a few trainable tokens, called soft prompts, instead of fine-tuning the entire model, but they can be sensitive to initialization. Prefix Tuning mitigates this by learning continuous vectors as prompts. Continuous vectors in Prefix Tuning are learnable parameters initialized in a high-dimensional space, whereas soft prompts are discrete token embeddings that depend heavily on specific initialization, making Prefix Tuning less prone to initialization sensitivity.

**Prefix Tuning** Li & Liang (2021) is a specialized form of prompt-based fine-tuning that focuses on prepending learnable continuous vectors, known as "prefixes," to the inputs. It involves the optimization of continuous vectors that shift the model towards specific downstream tasks. Prefix Tuning updates only the prefixes during fine-tuning, keeping the base model parameters frozen. This makes it significantly more memory efficient and scalable, especially for large-scale models. However, Prefix tuning remains sensitive to initialization, which may limit its adaptability in multitask settings.

**LoRA** Hu et al. (2021) reduces the number of trainable parameters by applying low-rank decomposition to simulate weight updates in frozen models, enabling efficient fine-tuning without increasing inference costs. Several variants have been proposed to further enhance its efficiency and applicability. **AdaLoRA** Zhang et al. (2023) leverages singular value decomposition (SVD) to prune less significant components, while **rsLoRA** Kalajdzievski (2023) introduces a scaling factor to stabilize the rank. **DoRA** Liu et al. (2024) implements dynamic optimization of LoRA parameters during training to improve adaptability across learning tasks. In the context of Stable Diffusion, **Yeh et al.** Yeh et al. (2024) proposed a unified LoRA framework that applies different combinations of LoRA methods for various tasks. **VeRA** Kopiczko et al. (2024) introduces scaling vectors that adjust pairs of frozen random matrices shared across layers, further optimizing parameter efficiency. Despite these advancements, LoRA and its variants are still primarily designed for single-task scenarios, with limited attention to multi-tasking environment.

**Multi-task learning (MTL)** trains models to solve multiple related tasks simultaneously by sharing parameters across tasks Zhang & Yang (2017); Ruder (2017). It often involves fine-tuning on several tasks before transferring knowledge to a new one Vu et al. (2020); Raffel et al. (2020); Aghajanyan et al. (2021). Building on the foundational PEFT techniques, recent innovations have proposed MTL-specific adaptations designed to minimize interference between tasks while maintaining parameter efficiency. One approach, **MPT** Wang et al. (2023d), learns a shared transferable prompt distilled from multiple task-specific prompts and applies multiplicative low-rank adaptations for efficient downstream task specialization. However, its major drawback is the need to pre-train individual teacher prompts for each source task before distilling knowledge into a shared prompt, which introduces significant computational overhead. **MTL-LoRA** Yang et al. (2025) extends the original LoRA framework by introducing task-adaptive parameters that preserve task-specific information and reduce interference in shared low-dimensional spaces, enhancing multi-task adaptation. Unlike standard LoRA, which merges adapters into the base model, MTL-LoRA requires task-specific routing during inference, resulting in added latency. **MultiLoRA** Wang et al. (2023c) addresses the limitations of LoRA's reliance on top singular vectors by horizontally scaling LoRA modules and diversifying their initialization, resulting in more balanced and effective adaptation across diverse tasks. However, it introduces a linear increase in VRAM usage during training due to the activation caching required for multiple parallel LoRA modules. **Customized Polytropon (C-Poly)** Wang et al. (2023a) is a modular, skill-based framework that enhances multi-task learning by combining shared and task-specific low-rank parameters through a learned skill assignment matrix. It struggles with unseen tasks, relying on it's fixed architecture with static skill modules limits generalization to novel tasks, reducing practicality in open-ended environments. Furthermore, existing approaches lack a mechanism to automatically adapt their architecture or hyperparameters to new or unseen data, requiring manual adjustments for each benchmarks.

## 3 PEML

PEML strategically handle the challenges of prompt alignment and low-rank model adaptability in multi-task learning by integrating standard LoRA with PrefixNAS. Unlike existing approaches that extend LoRA (such as MTL-LoRA, MultiLoRA) without considering for prompt alignment, PEML introduces a more cohesive framework where PrefixNAS dynamically adjusts the prefix structure based on the specific requirements of each task. This adaptability allows PEML to maintain

a task-responsive prefix module throughout training and inference, reducing the reliance on static architecture or any pre-trained teacher prefixes. Meanwhile, LoRA operates in parallel during training to effectively optimize low-rank adaption of the model without introducing a linear increase in VRAM usage. After training, LoRA is merged with the base model, resulting in a leaner architecture that only retains the adaptive PrefixNAS module. PrefixNAS is allowed to continuously refine its structure based on evolving task demands and ensuring optimal prompt alignment and model generalization.

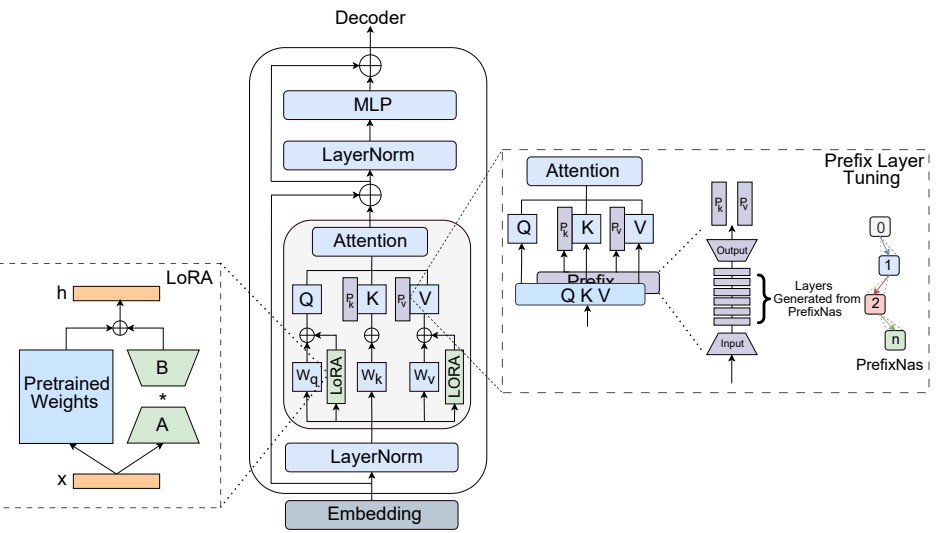

Figure 2: Unified view of PEML method. The left side illustrates LoRA, where matrix B is set to 0 and matrix A follows a normal distribution N(0, $\sigma^2$). The right side represents PrefixNAS, showcasing the optimal architecture derived from the search process.

PEML combines LoRA with PrefixNAS mechanism which use a gradient-based NAS approach that's built on continuous relaxation techniques. Both techniques optimize parallelly during training, as shown in Figure 2. This allows the model to adjust task-specific features and dynamically generate prefixes at the same time. A detailed breakdown of how this works is provided in Algorithm 1 and convergence analysis mentioned in 7.1. In Algorithm 1 the base model $\Phi$ is initialized with frozen pre-trained weights, while LoRA parameters $B$, $A$ and PrefixNAS architecture parameters $\alpha$ are trained concurrently. In each iteration, PrefixNAS generates candidate prefix architectures $\mathcal{A}_i(\alpha)$, which are used to construct task-specific prefixes $\mathbf{P}_i$. These prefixes are concatenated with inputs and passed through the model adapted by LoRA parameters, producing predictions and computing a combined loss consisting of task-specific and architecture regularization terms. LoRA and

---

**Algorithm 1** PEML: Joint Optimization with LoRA & PrefixNAS

$\Phi \leftarrow$ pre-trained model
$\theta \leftarrow$ base weights [Frozen]
$\{D_i\}_{i=1}^n \leftarrow$ datasets
$h \leftarrow$ PrefixNAS search space for prefixes
$s \leftarrow$ PrefixNAS operations
Initialize LoRA params $B$, $A$ and PrefixNAS params $\alpha$
**Parallel Joint Optimization**
**for** joint iteration $t = 1$ **to** $T$ **do**
    Generate prefix architectures: $\mathcal{A}_i(\alpha) \leftarrow$ PrefixNAS$(s)$
    $\mathbf{P}_i \leftarrow$ LearnablePrefix$(\mathcal{A}_i(\alpha), h)$ {NAS-optimized prefix}
    **for** each dataset $D_i$ **do**
        $\tilde{X}_i = \mathbf{P}_i \oplus X_i$ {Input with a prepend prefix}
        $\theta' \leftarrow \theta + B \cdot A$ {LoRA adaptation}
        Forward pass: $\hat{y} = \Phi(\tilde{X}_i; \theta')$
        Compute loss: $\mathcal{L}_t = \mathcal{L}(\hat{y}, y_i) + \lambda\mathcal{R}(\alpha)$
        **Simultaneous Updates:**
        ▷ Update LoRA params: $B, A \leftarrow B, A - \eta\nabla_{B,A}\mathcal{L}_t$
        ▷ Update NAS params: $\alpha \leftarrow \alpha - \eta\nabla_\alpha\mathcal{L}_t$
    **end for**
    Prune weak architectures via PrefixNAS
**end for**
**return** $\Phi_{\text{final}}(\theta', \alpha^*)$ {Jointly optimized model}

---

PrefixNAS parameters update via gradient descent, followed by pruning weak architectures. The final model $\Phi_{\text{final}}$ consists of the optimized LoRA weights and a unified Prefix architecture.

## 3.1 PROBLEM STATEMENT

Let $\mathcal{D} = \{D_1, D_2, \ldots, D_n\}$ denote multi-task datasets from different benchmarks, where each dataset $D_i = \{(x_{ij}, y_{ij})\}_{j=1}^{m_i}$ contains input-output pairs for task $T_i$. Mini-batches are constructed through parallel sampling:

$$B_k = \bigcup_{i=1}^{n} \{(x_{i1}, y_{i1}), \ldots, (x_{ib_i}, y_{ib_i})\} \quad \text{with} \quad b_i = \lfloor \gamma m_i \rfloor \tag{1}$$

The model $f_\theta$ with base parameters $\theta$ undergoes joint optimization through LoRA and PrefixNAS. LoRA modifies low-rank matrices $B, A \in \mathbb{R}^{d \times r}$ with $r \ll d$. The adapted parameter set $\theta'$ is expressed as:

$$\theta' = \theta + \Delta\theta = \theta + BA^\top \tag{2}$$

PrefixNAS employs a learnable prefix matrix $P \in \mathbb{R}^{l \times d}$, concatenated with the input sequence $X_i$. The transformed input $\tilde{X}_i$ is given by:

$$\tilde{X}i = \mathcal{A}_\alpha(P) \oplus X_i \tag{3}$$

where $\mathcal{A}_\alpha$ is an architecture search function parameterized by $\alpha$.

The joint loss function for PEML is defined as:

$$\mathcal{L}_{\text{joint}} = \frac{1}{n} \sum_{i=1}^{n} \frac{1}{|B_k^{(i)}|} \sum_{(x,y) \in B_k^{(i)}} \mathcal{L}(f_{\theta'}(\tilde{x}), y) + \lambda \mathcal{R}(\alpha) \tag{4}$$

Where the joint loss function $\mathcal{L}_{\text{joint}}$ combines task loss and architecture regularization. The task loss averages over mini-batches and samples, expressed as $\mathcal{L}(f_{\theta'}(\tilde{x}), y)$. The regularization term $\lambda \mathcal{R}(\alpha)$ applies a penalty to architecture parameters $\alpha$, scaled by $\lambda$.

During training, $\theta$ remains frozen, and the updates only $B$, $A$, and $\alpha$ alongside with $\mathcal{A}(P)$. After training, the final model integrated with PEML is expressed as:

$$f_{\theta_{\text{final}}} = f_{\theta + BA^\top} \circ \mathcal{A}_{\alpha^*}(P) \tag{5}$$

where $\alpha^*$ denotes the optimized architecture parameters obtained through PrefixNAS.

## 3.2 PREFIXNAS OPTIMIZATION

PrefixNAS generates adaptive architectures through differentiable search, enabling task-specific optimization beyond static embedding layers. For each task $T_i$, the prefix architecture combines candidate operations via continuous relaxation:

$$\mathcal{A}_i(\alpha_i) = \sum_{j=1}^{k} \frac{\exp(\alpha_{ij})}{\sum_{m=1}^{k} \exp(\alpha_{im})} \cdot o_j(P_i) \tag{6}$$

where $\alpha_i \in \mathbb{R}^k$ represents learnable architecture parameters, $P_i$ is the task-specific prefix, and $\{o_j\}_{j=1}^{k}$ denotes the set of candidate operations. After convergence, the final architecture is obtained by selecting the dominant operation:

$$\hat{\mathcal{A}}_i = o_{\operatorname{argmax}_j \alpha_{ij}} \tag{7}$$

PrefixNAS maintains differentiability during training while producing discrete, efficient architectures. Each task's prefix architecture is optimized independently, allowing specialized adaptation without inter-task interference.

## 3.3 PEML OPTIMIZATION

PEML integrated on a pre-trained model $\Phi(X; \theta)$ with frozen base parameters $\theta$. The adaptation combines low-rank updates $\Delta = BA^\top$ where $B, A \in \mathbb{R}^{d \times r}$, and task-specific prefixes $P_i$ generated through PrefixNAS with architecture parameters $\alpha$. The modified forward pass handle inputs as:

$$\tilde{X}_i = \mathcal{A}_\alpha(P_i) \oplus X_i, \quad \theta' = \theta + \Delta \tag{8}$$

The unified objective maximizes the log-likelihood with architectural regularization:

$$\max_{\Delta, \alpha} \sum_{i=1}^{n} \sum_{t=1}^{|Y_i|} \log p_{\theta'}(y_{i,t} | \tilde{X}_i, y_{i,<t}) - \lambda \mathcal{R}(\alpha) \tag{9}$$

### 3.4 HYPERPARAMETER OPTIMIZATION

PEML performs bi-level optimization where the inner loop optimizes the architectural parameters using PrefixNAS, and the outer loop optimizes hyperparameters through Tree-structured Parzen Estimator (TPE). Let $\mathbf{h} = \{h_1, ..., h_n\}$ represent the hyperparameters, and $\alpha = \{\alpha_i\}_{i=1}^{k}$ denote the PrefixNAS architecture parameters.

**Inner Loop (PrefixNAS Optimization)** searches for the optimal prefix architecture by evaluating multiple configurations and selecting those that maximize the objective function:

$$\alpha_t \sim p_t(\alpha) \tag{10}$$

For each sampled $\alpha_t$, the model is trained with the prefix defined by $\mathcal{A}_{\alpha_t}(P)$ and evaluated using:

$$f(\alpha_t) = \frac{1}{m} \sum_{j=1}^{m} A(\Phi(X_j; \theta' + \Delta, \mathcal{A}_{\alpha_t}(P))) \tag{11}$$

**Outer Loop (Hyperparameter Optimization with TPE)** samples hyperparameters and guiding the search based on previous evaluations for each trial $t$:

$$\mathbf{h}_t \sim p_t(\mathbf{h}) \tag{12}$$

The objective function for the outer loop becomes:

$$f(\mathbf{h}_t, \alpha^*) = \frac{1}{m} \sum_{j=1}^{m} A(\Phi(X_j; \theta' + \Delta, \mathcal{A}_{\alpha^*}(P), \mathbf{h}_t)) \tag{13}$$

TPE updates the sampling distributions for hyperparameters based on the evaluation results, while the architecture parameters are refined through PrefixNAS:

$$p_{t+1}(\mathbf{h}) \propto \exp(f(\mathbf{h}_t, \alpha^*)), \quad p_{t+1}(\alpha) \propto \exp(f(\alpha_t)) \tag{14}$$

The optimization process continues iteratively, refining both hyperparameters and prefix configurations to converge to the optimal combination $(\mathbf{h}^*, \alpha^*)$, defined as:

$$\mathbf{h}^*, \alpha^* = \underset{\mathbf{h}, \alpha}{\arg\max} f(\mathbf{h}, \alpha) \tag{15}$$

## 4 EXPERIMENTS

### 4.1 MODELS & DATASET

We evaluate PEML using T5-Large (770M) Raffel et al. (2020), FLAN-T5-Large Chung et al. (2024), LLaMA-7B Touvron et al. (2023) and LLaMA2-7B Touvron et al. (2023). Our experiments span across **GLUE** Wang et al. (2018) (SST-2 Socher et al. (2013), COLA Warstadt et al. (2018), STS-B Cer et al. (2017)), **SuperGLUE** Wang et al. (2019) (RTE Dagan et al. (2006), Boolq Clark et al. (2019), WIC Pilehvar & Camacho-Collados (2018)), **MMLU** Hendrycks et al. (2021), and commonsense reasoning tasks (PIQA Bisk et al. (2020), SIQA Sap et al. (2019), Winogrande Sakaguchi et al. (2020), OBQA Mihaylov et al. (2018), HellaSwag Zellers et al. (2019), ARC Clark et al. (2018)).

### 4.2 EXPERIMENTAL SETUP

In this study, we implemented various PEFT methods using the Hugging Face `PEFT` Mangrulkar et al. (2022) library, including PreEmbedd, PrefixNAS, and LoRA variants like DoRA and AdaLoRA. PreEmbedd consists only an embedding layer and an output layer, without any intermediate layers, virtual tokens is set to `20`, and the learning rate is fixed at `1e-3`. The LoRA and AdaLoRA configurations are as follows: rank `r = 16`, `LoRA_alpha = 32`, and `LoRA_dropout = 0.1`. We also provide huggingface modularity to PrefixNAS. It defines operations in the search space $O$ as linear transformations with dimensions of `(1024 x 1024)`. Each transformation is associated with an activation function (`ReLU, Tanh, Leaky ReLU, or GELU`), `dropout layers = [0.1, 0.3, or 0.5]`, and `layer normalization`. PrefixNAS generates `n = 6` layers between an embedding layer and a fixed output layer. TPE Watanabe (2023) is integrated with the PrefixNAS framework to refine hyperparameter search. The search required approximately 2 hours on 8× A100 GPUs for LLaMa variants (16 GPU-hours, ∼1.1 PFLOPs) and around 30 minutes

on 8× A100 GPUs for T5-large variants (4 GPU-hours, ∼0.28 PFLOPs). This search process is a one-time effort for each benchmark. The learning rate is sampled from a logarithmic uniform distribution between `0.001` and `0.02`, with a base step of `5e-5`. The prefix length varies from `5` to `50`. A total of `n=100` trials are conducted, each running for a maximum of `150 epochs`. The early stopping function is applied to terminate training after `25 epochs` without improvement in average accuracy. `Ray Tune` Liaw et al. (2018) framework is used to implement TPE. For efficient multi-GPU training, the training is distributed across `8 NVIDIA A100 40 GB GPUs` using huggingface `Accelerate` Gugger et al. (2022) library.

## 4.3 RESULTS

### 4.3.1 GENERAL LANGUAGE UNDERSTANDING

As shown in Table 1, PEML improves the average accuracy by 3.59% on GLUE benchmark compared to standalone LoRA, while it shows a smaller improvement of 0.71% over standalone AdaLoRA. Combining PreEmbedd with LoRA or AdaLoRA showed almost the same performance as using LoRA or AdaLoRA alone which proves our hypothesis that LoRA may limit the effectiveness of prompt alignment. PrefixNAS addresses this issue by optimizing the prefix architecture to better align prompts.

Table 1: Performance of various PEFT methods, including PreEmbedd, LoRA, AdaLoRA, and their combinations with PEML tested on the T5-large model across seven GLUE tasks.

| Peft Techniques | SST2 | MRPC | RTE | COLA | QQP | WNLI | QNLI | AVG |
|---|---|---|---|---|---|---|---|---|
| PreEmbedd | 0.895 | 0.799 | 0.893 | 0.753 | 0.962 | 0.734 | 0.879 | 0.845 |
| LoRA | 0.954 | 0.815 | **0.901** | 0.779 | 0.935 | 0.765 | 0.943 | 0.870 |
| LoRA-PreEmbedd | 0.966 | 0.851 | 0.891 | 0.834 | 0.961 | 0.671 | 0.965 | 0.877 |
| PEML | **0.976** | **0.867** | 0.899 | **0.843** | **0.971** | **0.781** | **0.974** | **0.901** |
| AdaLoRA | 0.951 | 0.891 | 0.910 | 0.880 | 0.960 | **0.812** | 0.962 | 0.908 |
| AdaLoRA-PreEmbedd | 0.960 | 0.883 | 0.912 | **0.890** | **0.973** | 0.7812 | 0.966 | 0.909 |
| PEML-AdaLoRA | **0.966** | **0.906** | **0.934** | 0.889 | 0.972 | 0.767 | **0.975** | **0.916** |

Table 2 presents a comparison of PEML against state-of-the-art multitask PEFT techniques on the GLUE benchmark using the LLaMA2-7B model. PEML achieves the highest average performance of 91.1%, outperforming all baselines. All experiments and the instructions prompt are follow the experimental setup described inYang et al. (2025).

Table 2: Compare the performance accuracy (COLA : mcc & STS-B: pea.) of PEML using LLaMA2-7B model with state-of-the-art multitasking framework.

| Peft Techniques | COLA | MNLI | MRPC | QNLI | QQP | RTE | SST2 | STSB | AVG |
|---|---|---|---|---|---|---|---|---|---|
| LoRA-MT | 0.659 | 0.914 | 0.860 | 0.960 | 0.909 | 0.917 | **0.971** | 0.919 | 0.889 |
| MultiLoRA | 0.613 | 0.910 | 0.863 | 0.955 | 0.900 | 0.910 | 0.963 | 0.918 | 0.879 |
| MoELoRA | 0.637 | 0.912 | 0.855 | 0.957 | 0.906 | 0.921 | 0.966 | 0.922 | 0.884 |
| MTL-LoRA | 0.680 | 0.914 | 0.902 | 0.963 | 0.914 | **0.924** | **0.971** | 0.928 | 0.900 |
| PEML | **0.698** | **0.964** | **0.912** | **0.967** | **0.928** | 0.912 | **0.971** | **0.935** | **0.911** |

### 4.3.2 MULTI-SENTENCE ADVANCED REASONING

Table 3 demonstrates the effectiveness of PEML across SuperGLUE benchmark. It achieves the highest overall average score of 88.08%, outperforming standalone PEFT baselines such as PreEmbedd (84.78%) and LoRA (83.67%) by +3.30% and +4.41%, respectively. Compared to standalone AdaLoRA, which achieves an average of 86.94%, PEML still provides a relative improvement of +1.14%. However, WSC is extremely sensitive to syntactic cues and entity disambiguation, which may benefit more from AdaLoRA's selective parameter tuning. In this case, pruning less useful LoRA components helps sharpen specific linguistic cues, and PrefixNAS provides minimal yet relevant contextual prompts leading to improved generalization.

Table 3: Performance of various PEFT methods, including PreEmbedd, LoRA, AdaLoRA, and their combinations with PEML tested on the T5-large model across SuperGLUE benchmark.

| Peft Techniques | Boolq | RTE | COPA | MultiRC | WIC | WSC | AVG |
|---|---|---|---|---|---|---|---|
| PreEmbedd | 0.918 | **0.939** | 0.775 | 0.897 | 0.841 | 0.714 | 0.847 |
| LoRA | 0.921 | 0.907 | 0.750 | 0.891 | 0.800 | **0.750** | 0.836 |
| LoRA-PreEmbedd | 0.926 | 0.931 | 0.750 | 0.902 | 0.830 | 0.678 | 0.836 |
| PEML | **0.934** | 0.9317 | **0.825** | **0.904** | **0.870** | **0.750** | **0.869** |
| AdaLoRA | **0.940** | **0.959** | 0.800 | **0.905** | 0.878 | 0.732 | 0.869 |
| AdaLoRA-PreEmbedd | 0.935 | 0.951 | **0.875** | 0.901 | **0.886** | 0.714 | 0.877 |
| PEML-AdaLoRA | 0.924 | 0.935 | 0.850 | 0.900 | 0.852 | **0.821** | **0.880** |

We also compare PEML with the state-of-the-art techniques in table 4 using the FLAN-T5-Large. PEML achieves the highest average accuracy of 84.31%, surpassing competitive baselines such as C-Poly (83.21%) and MOE-LoRA (82.31%) by +1.32% and +2.43%, respectively. Although C-Poly and PEML have similar performance, the C-Poly method struggles to handle unseen data due to its fixed architecture, while PrefixNAS offers a dynamic architecture that is better suited for handling unseen data.

Table 4: Compare the performance accuracy of PEML using FLAN-T5-Large model with state-of-the-art multitasking framework. All the results are directly reported from Wang et al. (2023a).

| Peft Techniques | Boolq | CB | COPA | MultiRC | RTE | WIC | WSC | AVG |
|---|---|---|---|---|---|---|---|---|
| LoRA | 0.818 | 0.857 | 0.900 | 0.827 | 0.859 | 0.595 | 0.644 | 0.818 |
| MOE-LoRA | 0.851 | **0.875** | **0.910** | 0.834 | 0.864 | 0.579 | 0.663 | 0.823 |
| Poly | 0.851 | **0.875** | 0.900 | 0.825 | 0.862 | 0.606 | **0.769** | 0.820 |
| MHR | 0.850 | **0.875** | 0.900 | 0.829 | 0.862 | 0.611 | 0.759 | 0.823 |
| C-Poly | 0.856 | 0.857 | 0.900 | 0.833 | 0.888 | 0.670 | 0.750 | 0.832 |
| PEML | **0.864** | 0.856 | 0.825 | **0.904** | **0.901** | **0.800** | 0.750 | **0.843** |

### 4.3.3 MASSIVE MULTITASK LANGUAGE UNDERSTANDING

We mixed four SuperGLUE tasks with MMLU in Table 5 and compared them with full fine-tuning (FT) and PEFT baselines. We demonstrates a comparable total rank budget across configurations. MultiLoRA employs three LoRA modules, each with a rank of 32 (total rank = $3 \times 32 = 96$), while PEML uses a single LoRA module with rank 96. Consequently, both configurations share the same overall rank budget. PEML achieves the highest average accuracy of 80.3%, outperforming the FT (76.9%) by +3.4% and the most computationally intensive MultiLoRA configuration (n=5, r=32) by +2.3%. It produces similar results but at the cost of increased VRAM usage as n scales ($\sim$ see figure 4). However, our method does not require horizontal scaling, and the results support that PEML is more efficient than MultiLoRA in terms of performance and resource optimization.

Table 5: Compare the performance accuracy of PEML using LLaMA-7B model with state-of-the-art multitasking framework. MMLU is tested with 5-shot prompts and SuperGLUE are tested with zero-shot. All results are reported directly from Wang et al. (2023c).

| Peft Techniques | MMLU | Boolq | MultiRC | RTE | WIC | AVG |
|---|---|---|---|---|---|---|
| FT | 0.495 | 0.884 | 0.872 | 0.852 | 0.740 | 0.769 |
| $\text{LoRA}_{r=96}^{n=1}$ | 0.477 | 0.882 | 0.854 | 0.834 | 0.716 | 0.752 |
| $\text{LoRA}_{r=160}^{n=1}$ | 0.502 | 0.877 | 0.853 | 0.833 | 0.701 | 0.753 |
| $\text{MultiLoRA}_{r=32}^{n=3}$ | 0.512 | 0.878 | 0.887 | **0.897** | 0.708 | 0.776 |
| $\text{MultiLoRA}_{r=32}^{n=5}$ | 0.514 | 0.885 | 0.894 | 0.894 | 0.714 | 0.780 |
| $\text{PEML}_{r=96}^{n=1}$ | **0.516** | **0.923** | **0.928** | 0.896 | **0.755** | **0.803** |

### 4.3.4 COMMONSENSE REASONING

Table 6 presents performance comparison across eight commonsense reasoning tasks using various PEFT techniques. All the instructions prompt are same as Liu et al. (2024); Yang et al. (2025). PEML outperforming DoRA (80.5%) and MoELoRA (78.3%) by +2.52% and +4.72%, respectively. DoRA, like other LoRA variants, ignores prompt alignment and MoE needs to store many experts for different tasks. It adds extra latency when switching experts. However, in PEML, we have a unified architecture, so there's no need to switch adapters.

Table 6: Commonsense reasoning results on LLaMA2-7B. We follow the joint training setup described in Liu et al. (2024); Yang et al. (2025), and all results are reported directly from those works.

| Peft Techniques | Boolq | PIQA | SIQA | Winogrande | OBQA | Hellaswag | ARC-E | ARC-C | AVG |
|---|---|---|---|---|---|---|---|---|---|
| LoRA | 0.698 | 0.799 | 0.795 | 0.826 | 0.810 | 0.836 | 0.798 | 0.647 | 0.776 |
| DoRA | 0.720 | 0.831 | 0.799 | 0.830 | 0.812 | 0.891 | 0.845 | 0.710 | 0.805 |
| MultiLoRA | 0.665 | 0.658 | 0.628 | 0.793 | 0.754 | 0.792 | 0.767 | 0.596 | 0.707 |
| MoELoRA | 0.680 | 0.835 | 0.704 | 0.825 | 0.832 | 0.906 | 0.868 | 0.615 | 0.783 |
| MTL-LoRA | 0.710 | 0.844 | 0.808 | 0.849 | 0.826 | **0.931** | 0.870 | 0.734 | 0.821 |
| PEML | **0.775** | **0.887** | **0.837** | **0.902** | **0.838** | 0.774 | **0.885** | **0.741** | **0.830** |

## 5 SENSITIVITY ANALYSIS

In this section, we examine the robustness of PrefixNAS across various settings, focusing on the number of layers (n), repetition of blocks (b), and the inclusion of skip connections (sc) and reduction cells (rc). Using T5-large as the backbone model, we conduct our analysis on the SuperGLUE benchmark. As shown in Figure 3, our findings indicate that setting n=6 consistently delivers optimal performance. Repeating the same block multiple times does not significantly impact the results, while the inclusion of skip connections and reduction cells appears to limit further performance gains, suggesting that further structural changes offer minimal gains. Therefore, We exclude them from the NAS search operation, thereby reducing complexity and potentially leading to faster convergence.

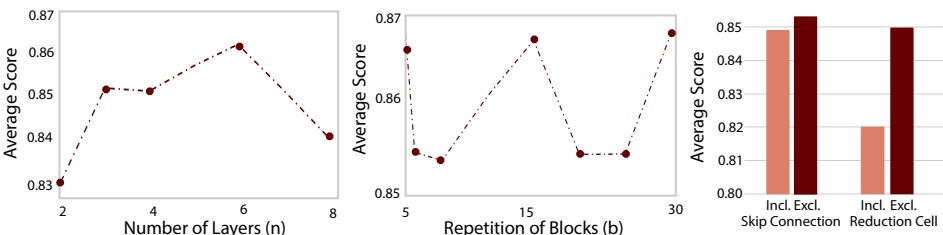

Figure 3: The performance of PEML on SuperGLUE benchmark with different sensitive hyperparameter configurations.

## 6 CONCLUSION

In this research, we introduced PEML, a novel approach designed to overcome the limitations of existing state-of-the-art (SOTA) PEFT methods in multi-task learning. SOTA techniques such as MTL, MultiLoRA, C-Poly, and MoE often struggle with challenges such as prompt misalignment, static task specific architecture, multiple adapter switching during inference and linear increase in VRAM during training. PEML effectively addresses these issues by dynamically select the optimize prefix architecture through PrefixNAS. PEML reduces the need for multiple adapters as it is a unified structure. One of the limitations of PEML introduces additional parameters to the prefix in order to unify multiple tasks, and also incurs resource cost during NAS search process. Our experimental results illustrate that PEML achieves an average accuracy improvement of up to 6.67% across various tasks, with some individual tasks showing enhancements of up to 10.75%. These results highlight the capability of PEML to enhance multi-task learning in natural language understanding.

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

# 7 APPENDIX

## 7.1 CONVERGENCE ANALYSIS

This section analyzes the convergence properties of PEML, which jointly optimizes LoRA and PrefixNAS. The analysis considers the coupled dynamics of LoRA parameters, Prefix parameters, and PrefixNAS search variables under a unified optimization framework. The objective is to minimize the function $f(\theta_{\text{LoRA}}, \theta_{\text{Prefix}}, \alpha)$, where $\alpha$ determines the selection of operations in the search process. The parameters are updated simultaneously using stochastic gradients, with $\alpha$ constrained to a simple $\sum_i \alpha_i = 1, \alpha_i \geq 0$. The problem formulation involves three parameter groups: the LoRA parameters ($\theta_{\text{LoRA}}$), the Prefix parameters ($\theta_{\text{Prefix}}$), and the architecture weights ($\alpha$). The joint optimization objective can be expressed as:

$$\min_{\theta \in \Theta} f(\theta) \quad [\text{where} \quad \theta = \theta_{\text{LoRA}}, \theta_{\text{Prefix}}, \alpha] \tag{16}$$

The convergence analysis is based on several key assumptions. (1) The loss function $f$ is assumed to be $\beta$-smooth in all parameters:

$$|\nabla f(\theta) - \nabla f(\theta')| \leq \beta |\theta - \theta'| \tag{17}$$

(2) The gradients of the parameters are bounded:

$$|\nabla_{\theta_{\text{LoRA}}} f| \leq M_L, \quad |\nabla_{\theta_{\text{Prefix}}, \alpha]} f| \leq M_P \tag{18}$$

where $M = \max(M_L, M_P)$ prevents gradient explosion in any module.

(3) A unified learning rate $\eta_t = \frac{c}{\sqrt{T}}$ with effective step sizes defined for each parameter group as follows:

$$\eta_t^{\text{LoRA}} = \eta_t^{\text{Prefix}} = \eta_t^{\alpha} = \eta_t \tag{19}$$

The optimization dynamics involve projected SGD for NAS variables and standard SGD for LoRA and Prefix parameters:

$$\theta_{\text{LoRA}}^{t+1} = \theta_{\text{LoRA}}^t - \eta_t \nabla_{\theta_{\text{LoRA}}} f(\theta^t)$$
$$[\theta_{\text{Prefix}}^{t+1}, \alpha^{t+1}] = \Pi_{\mathcal{K}} \left( [\theta_{\text{Prefix}}^t, \alpha^t] - \eta_t \nabla_{[\theta_{\text{Prefix}}, \alpha]} f(\theta^t) \right) \tag{20}$$

Applying the descent lemma under the $\beta$-smoothness assumption, the per-iteration loss change can be expressed as:

$$f(\theta^{t+1}) \leq f(\theta^t) - \eta_t \left( |\nabla_{\theta_{\text{LoRA}}} f(\theta^t)|^2 + |\nabla_{\theta_{\text{Prefix}}, \alpha]} f(\theta^t)|^2 \right)$$
$$+ \frac{\beta \eta_t^2}{2} \left( M_L^2 + M_P^2 \right) \tag{21}$$

Summing over $T$ iterations and utilizing the gradient bounds, the convergence bound can be derived as:

$$\frac{1}{T} \sum_{t=0}^{T-1} \left( |\nabla_{\theta_{\text{LoRA}}} f(\theta^t)|^2 + |\nabla_{\theta_{\text{Prefix}}, \alpha]} f(\theta^t)|^2 \right) \leq \frac{C(M_L + M_P)}{\sqrt{T}} \tag{22}$$

where $C$ is a constant that depends on $\beta$, $c$, and $c_\alpha$.

Under the specified assumptions and a learning rate schedule of $\eta_t = \frac{c}{\sqrt{T}}$, the final convergence guarantee after $T$ iterations is given by:

$$\frac{1}{T}\sum_{t=0}^{T-1}\mathbb{E}[|\nabla f(\theta^t)|^2] \le \frac{2(f(\theta^0) - f^*) + \beta c^2(M_L^2 + M_P^2)}{c\sqrt{T}} \tag{23}$$

where the gradient norm incorporates contributions from both parameter optimization and architecture search. The analysis ensures stable updates across LoRA, Prefix Tuning, and NAS variables under a unified learning rate framework.

## 7.2 ADDITIONAL RESULTS

### 7.2.1 PERFORMANCE ANALYSIS ON LOW DATA SETTINGS

We also evaluated cross-task knowledge transfer by training on a small sample from all tasks, observing comparable results as shown in Table 7 and Table 8. We evaluated the performance of various PEFT techniques, including PreEmbedd, LoRA, and AdaLoRA, along with their combinations with PEML, on GLUE and SuperGLUE benchmarks. PEML achieved the highest average accuracy with LoRA and AdaLoRA on both benchmark in resource-constrained scenarios.

Table 7: Performance of various PEFT methods, including PreEmbedd, LoRA, AdaLoRA, and their combinations with PEML, tested on the T5-large model on GLUE benchmark. Results are reported for low data (500 samples from each task) settings.

| Peft Techniques | SST2 | MRPC | RTE | COLA | QQP | WNLI | QNLI | AVG |
|---|---|---|---|---|---|---|---|---|
| PreEmbedd | 0.855 | 0.811 | 0.888 | 0.806 | 0.936 | 0.687 | 0.856 | 0.834 |
| LoRA | 0.932 | 0.871 | 0.891 | 0.859 | 0.944 | 0.718 | 0.940 | 0.880 |
| LoRA & PreEmbedd | 0.938 | 0.863 | 0.910 | 0.875 | 0.944 | 0.718 | 0.939 | 0.884 |
| PEML | 0.952 | 0.932 | 0.942 | 0.890 | 0.942 | 0.750 | 0.952 | 0.908 |
| AdaLoRA | 0.932 | 0.895 | 0.896 | 0.871 | 0.950 | 0.734 | 0.952 | 0.890 |
| AdaLoRA & PreEmbedd | 0.916 | 0.891 | 0.893 | 0.846 | 0.945 | 0.750 | 0.946 | 0.884 |
| PEML-AdaLoRA | 0.968 | 0.907 | 0.920 | 0.835 | 0.972 | 0.796 | 0.962 | 0.904 |

Table 8: performance of various PEFT methods, including PreEmbedd, LoRA, AdaLoRA, and their combinations with PEML tested on the T5-large model on SuperGLUE tasks. Results are reported for low data (300 samples from each task) efficiency settings, with the PEML achieving the highest average performance.

| Peft Techniques | Boolq | RTE | COPA | MultiRC | WIC | WSC | AVG |
|---|---|---|---|---|---|---|---|
| PreEmbedd | 0.911 | 0.891 | 0.750 | 0.877 | 0.781 | 0.875 | 0.847 |
| LoRA | 0.919 | 0.935 | 0.750 | 0.889 | 0.810 | 0.732 | 0.839 |
| LoRA & PreEmbedd | 0.922 | 0.931 | 0.800 | 0.890 | 0.814 | 0.714 | 0.845 |
| PEML | 0.912 | 0.871 | 0.925 | 0.876 | 0.829 | 0.767 | 0.863 |
| AdaLoRA | 0.912 | 0.907 | 0.750 | 0.871 | 0.785 | 0.767 | 0.832 |
| AdaLoRA & PreEmbedd | 0.897 | 0.895 | 0.725 | 0.878 | 0.791 | 0.767 | 0.826 |
| PEML-AdaLoRA | 0.922 | 0.943 | 0.825 | 0.893 | 0.840 | 0.803 | 0.871 |

### 7.2.2 PERFORMANCE ANALYSIS ON GLM-10B

As detailed in Table 9, we also benchmarked PEML against other PEFT methods on the GLM-10B model. Our approach demonstrates clear superiority by achieving the highest accuracy on SuperGLUE benchmark with an average score of 0.631.

### 7.2.3 BENCHMARK COMPARISON: DEEPSEEK 7B VS QWEN2-7B VS LLAMA3-8B

We selected our primary models to ensure fair comparison with prior work, since they are widely used and well-established in the parameter-efficient tuning literature. Most baseline studies rely on these models, making them natural choices for consistent evaluation. While newer models such

Table 9: Compare the performance accuracy of PEML using GLM-10B Du et al. (2021) model with state-of-the-art multitasking framework. All the results are directly reported from Wang et al. (2023a).

| Peft Techniques | Boolq | CB | COPA | MultiRC | RTE | WIC | WSC | AVG |
|---|---|---|---|---|---|---|---|---|
| LoRA | 0.609 | 0.463 | 0.657 | 0.624 | 0.573 | 0.391 | 0.323 | 0.520 |
| MOE-LoRA | 0.633 | 0.450 | 0.634 | 0.640 | 0.612 | 0.403 | 0.396 | 0.538 |
| Poly | 0.646 | 0.521 | 0.655 | 0.656 | 0.621 | 0.417 | 0.4708 | 0.569 |
| MHR | 0.648 | 0.507 | 0.663 | 0.657 | 0.627 | 0.423 | 0.455 | 0.569 |
| C-Poly | 0.673 | 0.603 | 0.704 | 0.679 | 0.680 | 0.487 | 0.534 | 0.622 |
| PEML | **0.682** | **0.610** | **0.707** | **0.688** | **0.693** | **0.491** | **0.552** | **0.631** |

as LLaMa3-8B Grattafiori et al. (2024) , DeepSeek-LLM-7B Bi et al. (2024), and Qwen2-7B Bai et al. (2023) show strong potential, we opted for LLaMa2-7B due to its maturity, stable training dynamics, and broad community support. Nonetheless, we also evaluated our PEML approach on several state-of-the-art models across GLUE, SuperGLUE, MMLU, and Commonsense Reasoning (CR) benchmarks in table 10.

Table 10: Benchmark performance of recent LLMs on MMLU, GLUE, SuperGLUE, and Commonsense Reasoning (CR). Average accuracy across all benchmarks is reported for each model.

| SOTA Models | MMLU | GLUE | SuperGLUE | CR |
|---|---|---|---|---|
| LLaMA3-8B | 0.662 | 0.922 | 0.915 | 0.821 |
| DeepSeek-7B | 0.569 | 0.910 | 0.897 | 0.834 |
| Qwen2-7B | 0.694 | 0.904 | 0.882 | 0.809 |

## 7.3 ADDITIONAL PARAMETERS

PEML introduces a dynamic architecture search mechanism where the number of additional parameters is not predetermined but instead depends on the trajectory of the search and the sub-architectures selected during training. In practice, the overhead remains relatively small, typically within 2% to 8% of the base model's parameters.

## 7.4 ABLATION STUDY

We conduct ablation studies to evaluate the impact of three critical aspects in PEML: the optimization order in PEML the number of layers (n), and the search space operations within PrefixNAS. For each experiment, we use T5-large model and train it on the SuperGLUE dataset. (1) We observe that parallel optimization consistently outperforms sequential optimization—whether LoRA is followed by PrefixNAS or vice versa. Moreover, sequential order has more additional training overhead than the parallel order ($\sim$ see Appendix 7.6). (2) We increase the number of layers up to 8 but beyond n=6 does not yield a significant performance boost ($\sim$ see Figure 3). However, A more complex architecture with additional layers could theoretically improve performance but it also introduces a large number of trainable parameters. Therefore, we must find a balance between architectural complexity and the associated computational cost. Lastly, (3) We find that certain operations did not have any major contribution to the PrefixNAS architecture. We exclude these operations during our search space design ($\sim$ see Figure 3). This reduction in search space results in significant time savings during the search process.

## 7.5 PEML VS. MULTILORA: VRAM-EFFICIENT SCALING

Training throughput, and VRAM usage are critical for generative LLMs. MultiLoRA increases model capacity by adding multiple parallel LoRA modules ($n > 1$), which linearly increases VRAM usage due to activation caching, especially for long sequences. For example, training LLaMA-7B with sequences of 1024 tokens and n=5 modules can consume more VRAM than full-parameter fine-tuning, limiting practical scalability. In contrast, our proposed PEML maintains a single LoRA module (n=1) and increases the rank r to match or exceed the expressivity of MultiLoRA. This design

avoids horizontal scaling entirely and VRAM usage remains nearly constant regardless of the rank increase. Additionally, MultiLoRA and PEML do not introduce notable latency and the throughput remains close to around 400 tokens per GPU per second ($\sim$ see figure 4). In our benchmarking, the throughput of PEML is almost twice that of full parameter fine-tuning (208 tokens per GPU per second)

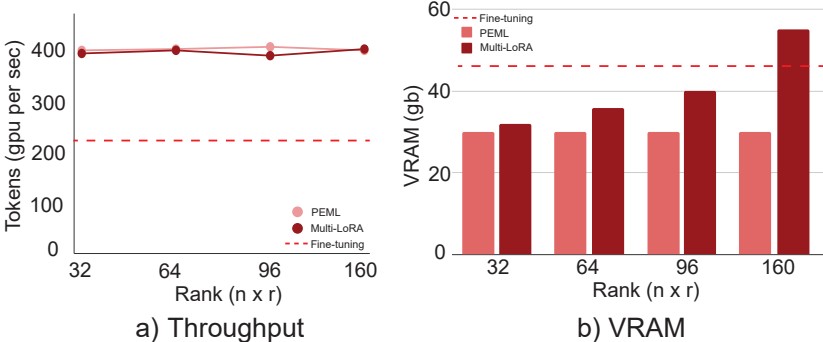

a) Throughput           b) VRAM

Figure 4: (a) Throughput and (b) peak VRAM usage benchmarked when training LLaMA-7B with sequences of 1024 tokens. n × r on horizontal axis indicates total rank of MultiLoRA and PEML.

## 7.6 COMPARISON OF SEQUENTIAL AND PARALLEL OPTIMIZATION

We further investigate the effect of combining LoRA and Prefix Tuning under different optimization approaches. Table 11 reports the performance on five representative SuperGLUE tasks. We consider three settings: (i) sequential optimization where LoRA is trained first followed by Prefix Tuning, (ii) sequential optimization in reverse order where Prefix Tuning is trained first followed by LoRA, and (iii) parallel optimization where LoRA and Prefix Tuning are jointly optimized (PEML). Results demonstrate that parallel optimization consistently yields higher average performance, while sequential variants show task-specific variations.

Table 11: Performance comparison of sequential and parallel optimization of LoRA and Prefix Tuning across SuperGLUE tasks. Parallel optimization achieves the highest average performance.

| Approches | BoolQ | COPA | RTE | WSC | WiC | Avg |
|---|---|---|---|---|---|---|
| LoRA → Prefix | 0.929 | 0.855 | 0.922 | 0.750 | 0.835 | 0.858 |
| Prefix → LoRA | 0.922 | 0.835 | 0.938 | 0.735 | 0.864 | 0.859 |
| LoRA ‖ Prefix (PEML) | 0.925 | 0.850 | 0.932 | 0.804 | 0.837 | **0.869** |

## 7.7 INFERENCE LATENCY INDUCED BY ADAPTER SWITCHING

Inference latency in multi-task setting is affected by PEFT adapter switching. For 100 tasks with separate adapters, each task's latency is the forward pass $t_f$ plus switch time $t_s$, giving total latency $T = 100 \times (t_f + t_s)$. In PEML, a single unified adapter removes switching ($t_s = 0$), so $T_{\text{PEML}} = 100 \times t_f$. Using empirical estimates with minor variability, T5-large has $t_f = 11$ ms, average $t_s \approx 2.1$ ms, resulting in 1,320 ms for 100 adapters vs 1,100 ms with PEML ($\sim$ 17% reduction); LLaMA2-7B has $t_f = 52$ ms, average $t_s \approx 4.3$ ms, giving 5,630 ms vs 5,200 ms ($\sim$ 8% reduction). The switching overhead scales linearly with the number of tasks, so PEML's unified-adapter design becomes increasingly advantageous for larger multi-task setups, maintaining efficiency without compromising forward-pass computation.

## 7.8 LLM USAGE

We used a large language model (LLM) solely for writing polish, such as improving grammar, clarity, and readability of the text. The LLM did not contribute to research ideation, methodology, analysis, or results. All scientific content and conclusions are the responsibility of the authors.

