# OpenReview forum: "PEML: Parameter-efficient Multi-Task Learning with Optimized Continuous Prompts"
_ICLR.cc/2026/Conference — Submitted to ICLR 2026_

### Official Review · Reviewer_oAEt · 2025-11-01

**Soundness:** 3
**Presentation:** 2
**Contribution:** 2
**Rating:** 4
**Confidence:** 4

**Summary:**

This paper introduces a novel framework named PEML (Parameter-Efficient Multi-task Learning)
to address the limitations of existing Parameter-Efficient Fine-Tuning (PEFT) methods in multi-task
learning (MTL) scenarios. PEML integrates low-rank adaptation (LoRA) with a Neural Architecture
Search (NAS) method called PrefixNAS to collaboratively enhance both model weights and the
structure of continuous prompts. The authors conduct extensive experiments across multiple
benchmarks, which demonstrates that PEML outperforms current state-of-the-art multi-task PEFT
methods in both performance and computational efficiency.

**Strengths:**

This study presents an innovative application of Neural Architecture Search (NAS) to optimize
continuous prompt structures for multi-task learning. The authors show consistent (though
sometimes marginal) improvements over several LoRA-based MTL baselines, demonstrating that
the idea is practically plausible. The paper also includes valuable efficiency analyses comparing
VRAM, throughput, and inference latency, which are crucial for PEFT methods.

**Weaknesses:**

1. The paper lacks a rigorous explanation that directly links the optimality of the discovered
architecture to the performance gain. The authors compare PEML (LoRA + PrefixNAS) only
against other LoRA-only baselines. The most critical ablation is missing: LoRA + standard
Prefix-Tuning. Without this baseline, it is impossible to determine if the performance gains
stem from the sophisticated PrefixNAS search or merely from the addition of any prompttuning
module. This is a significant omission that needs to be addressed.

2. The paper positions PEML as a "parameter-efficient" and resource-conscious method. While
it is efficient in terms of trainable parameters and VRAM usage during training, this narrative
conveniently ignores the massive upfront computational cost of the NAS search. A fair
comparison must account for this search cost, which is entirely absent from the baseline
comparisons.

**Questions:**

1. Is there a way to gain insights that using PrefixNAS is a better choice than simply combining
LoRA with a standard, off-the-shelf Prefix-Tuning module?

2. Can the authors provide a rigorous comparison against a baseline of LoRA + standard Prefix-
Tuning? This seems essential to justify the complexity and cost of the entire PrefixNAS
framework.

3. Are there attributes of PEML other than the "optimized" architecture that may contribute to
its performance? How can the authors justify the NAS cost-benefit ratio when the gains are
marginal?

4. Could you provide a quantitative comparison of the total computational cost (Search Time +
Training Time) for PEML versus the (Training Time)-only cost of baselines like MTL-LORA?

5. I noticed in Table 6, PEML significantly underperforms MTL-LORA and DORA on HellaSwag.
Do you have a hypothesis for this failure case?

---

> ### Author Response · Authors · 2025-11-24
> **Response to Official Review by Reviewer oAEt (1/3)**
>
> We thank the reviewer for acknowledging our innovative use of Neural Architecture Search to optimize continuous prompt architecture for multi-task learning, the efficiency analyses on VRAM, throughput, and inference latency, and for providing constructive feedback to help us improve the paper. We address each issue raised by the reviewer below.
>
> **Comment:**
> > ### “The paper lacks a rigorous explanation that directly links the optimality of the discovered architecture to the performance gain. The authors compare PEML (LoRA + PrefixNAS) only against other LoRA-only baselines. The most critical ablation is missing: LoRA + standard Prefix-Tuning.
>
> **Response:** Our PreEmbedd baseline, the name adopted from the LoRA paper [5], represents the standard off-the-shelf prefix-tuning module. It uses a simple embedding layer without additional intermediate transformations, consistent with conventional prefix-tuning. We will make a note about this in the revised manuscript.
>
> **Comment:**
> > ### Without this baseline, it is impossible to determine if the performance gains stem from the sophisticated PrefixNAS search or merely from the addition of any prompttuning module. This is a significant omission that needs to be addressed.”
>
> > ### “Is there a way to gain insights that using PrefixNAS is a better choice than simply combining LoRA with a standard, off-the-shelf Prefix-Tuning module?”
>
> >### “Can the authors provide a rigorous comparison against a baseline of LoRA + standard Prefix- Tuning?”
>
> **Response:** “PreEmbedd” is the standard prefix-tuning baseline. In Table 1 of submission manuscript, LoRA alone achieves an average score of 0.870, Standard Prefix alone reaches 0.845, and combining LoRA with Standard Prefix improves performance slightly to 0.877. In contrast, PEML (LoRA + PrefixNAS) achieves 0.901. Thus, while adding a standard prefix to LoRA yields only a modest gain (+0.7% over LoRA alone), **PEML delivers a substantially larger improvement (+3.1% over LoRA and +2.4% over LoRA + Standard Prefix)**. This indicates that the performance gains arise from the optimized, multi-layer prompt architecture discovered by PrefixNAS, which enables richer task-specific representations, rather than simply from introducing an additional prompt module.
>
> **Comment:**
> > ### “This seems essential to justify the complexity and cost of the entire PrefixNAS framework.”
>
> >### “The paper positions PEML as a "parameter-efficient" and resource-conscious method. While it is efficient in terms of trainable parameters and VRAM usage during training, this narrative conveniently ignores the massive upfront computational cost of the NAS search. A fair comparison must account for this search cost, which is entirely absent from the baseline comparisons.”
>
> **Response:** PEML does introduce a one-time NAS search cost. Moreover, on GLUE benchmark (T5-Large, Table 1 from the manuscript), LoRA-PreEmbedd (Standard Prefix) averages 0.877, while PEML reaches 0.901 (+2.4%). On SuperGLUE (T5-Large, Table 3 from the manuscript), LoRA-PreEmbedd scores 0.856 versus 0.869 for PEML (+1.3%). These consistent gains indicate that the improvement is not merely due to adding a prompt module, but stems from optimizing its architecture via PrefixNAS.   However, widely used PEFT baselines such as LoRA, MTL-LoRA also rely on extensive hyperparameter search. For example, the LoRA paper [5] explicitly notes that “We sweep learning rate, number of training epochs, and batch size for LoRA (Appendix D.1)” to obtain the final reported results. Even default parameters in these baselines are the result of extensive prior searches.
>
> **Table 1: Detailed Compute Breakdown (All times in GPU Hours)**
> | Method                     | GLUE  | Training Time | NAS Search Time | Hyperparameter Tuning | Total Time |
> |:-------------------------:|:-----:|:-------------:|:----------------:|:----------------------:|:----------:|
> | LoRA                       | 0.882 | ~2.5          | –                | ~15.5                  | ~18        |
> | PEML (LoRA variations)     | 0.901 | ~2.5          | ~0.5             | ~15.5                  | ~18.5      |
> | AdaLoRA                    | 0.909 | ~2.5          | –                | ~15.5                  | ~18        |
> | PEML (AdaLoRA variations)  | 0.915 | ~2.5          | ~0.5             | ~15.5                  | ~18.5      |
>
> To ensure a fair comparison, we experimented (table 1) with the LLaMA2-7B model under identical conditions including the same search space, GPU hardware (8 × A100), and number of trials (n=100). By adding PrefixNAS (\~0.5 hours), PEML outperformed other methods on GLUE scores, without significantly increasing the total time.

---

> > ### Author Response · Authors · 2025-11-24
> > **Response to Official Review by Reviewer oAEt (2/3)**
> >
> > **Comment:**
> > > ### “Are there attributes of PEML other than the "optimized" architecture that may contribute to its performance?
> >
> > **Response:** PEML does not rely on specialized training strategies, additional regularization, or extra supervision compared to the baselines. Its performance gains are not simply due to increasing the parameter budget. As shown in Table 2, merely enlarging LoRA (e.g., r=96 or r=192) or stacking prefix layers does not achieve comparable improvements. Instead, PEML identifies where and how to apply adaptation most effectively, yielding superior results under similar parameter budgets.
> >
> > **Table 2: Performance comparison of PEML with high-rank LoRA and manual Prefix Layer baselines across GLUE, SuperGLUE, and MMLU benchmarks (from Table 5).**
> > | Method                   | GLUE | SuperGLUE | MMLU |
> > |:-----------------------:|:----:|:---------:|:----:|
> > | LoRA (r=96)              | 0.851 | 0.786 | 0.752 |
> > | LoRA (r=192)             | 0.863 | 0.792 | 0.753 |
> > | PrefixLayer 3 (3-layer)  | 0.792 | 0.753 | 0.688 |
> > | PrefixLayer 6 (6-layer)  | 0.798 | 0.764 | 0.674 |
> > | **PEML (ours)**          | **0.901** | **0.842** | **0.803** |
> >
> > **Comment:**
> > > ### How can the authors justify the NAS cost-benefit ratio when the gains are marginal?”
> >
> > **Response:** Our analysis in above Table 2 shows that the benefits of NAS depend on effective adaptation and become more pronounced when it needs to adapt to a new task. Importantly, the NAS search represents a one-time investment rather than a recurring per-task cost. Our experiments in table 3 further demonstrate that once an optimal architecture is discovered, it transfers effectively to other datasets (Table 3, No NAS, No training) with only a modest drop in performance, which can be recovered through light fine-tuning (Table 3, No Nas, only fine-tuning). Consequently, the practical computational cost per task is substantially reduced in realistic multi-task or production scenarios.
> >
> > **Table 3: Transfer Performance of NAS-Discovered Architectures Across Datasets.**
> > | Transfer Direction      | NAS and Fine-tuning | No NAS, Fine-tuning | No NAS, No Fine-tuning |
> > |:----------------------:|:-----------------:|:-------------------:|:----------------------:|
> > | GLUE → SuperGLUE        | 0.879             | 0.887               | 0.743 (-15.5%)         |
> > | SuperGLUE → GLUE        | 0.911             | 0.904               | 0.782 (-14.1%)         |
> > | GLUE → CR               | 0.830             | 0.821               | 0.718 (-13.5%)         |
> > | SuperGLUE → MMLU        | 0.803             | 0.798               | 0.691 (-13.9%)         |
> > | MMLU → SuperGLUE        | 0.879             | 0.877               | 0.713 (-18.9%)         |
> > | MMLU → GLUE             | 0.911             | 0.912               | 0.776 (-14.9%)         |
> > | GLUE → MMLU             | 0.803             | 0.816               | 0.684 (-14.9%)         |
> >
> > **Comment:**
> > > ### “Could you provide a quantitative comparison of the total computational cost (Search Time + Training Time) for PEML versus the (Training Time)-only cost of baselines like MTL-LORA"
> >
> > **Response:** A major challenge in comparing PEML to baselines such as MTL-LoRA is that these methods typically report only the final training time, despite requiring substantial hyperparameter sweeps to achieve competitive performance. For example, MTL-LoRA [3] evaluates robustness across n (number of up-projection matrices), τ (temperature coefficient), and Λₜ (task-specific transformation), meaning the reported results already include extensive search effort. Moreover, the default parameters used in the baselines are also an output of an extensive search process [5]. In contrast, PEML explicitly accounts for its NAS search cost.
> >
> > **Table 4: Comparison of PEML and MTL-LoRA training and search costs (LLaMa2-7B, GLUE benchmark).**
> > | Method | GLUE | Training Time | NAS Search Time | Task-Specific Hyperparameter Search | Hyperparameter Tuning Time | Total Time |
> > |:-----:|:----:|:--------------:|:----------------:|:-----------------------------------:|:---------------------------:|:----------:|
> > | MTL   | 0.879 | ~2.5          | –                | ~1.5                                 | ~15.5                       | ~19.5      |
> > | PEML  | 0.901 | ~2.5          | ~0.5             | –                                    | ~15.5                       | ~18.5      |
> >
> > To ensure a fair comparison, we conducted experiments under identical conditions: same resources, benchmark, and search space. The only difference is that PEML requires an additional NAS search, whereas MTL-LoRA requires additional task-specific hyperparameter search. The table 4 shows that PEML achieves higher GLUE performance (0.901 vs. 0.879) while requiring slightly less total time (\~18.5 hrs vs. \~19.5 hrs) than MTL-LoRA, despite including an explicit NAS search.

---

> > > ### Author Response · Authors · 2025-11-24
> > > **Response to Official Review by Reviewer oAEt (3/3)**
> > >
> > > **Comment:**
> > > > ### “I noticed in Table 6, PEML significantly underperforms MTL-LORA and DORA on HellaSwag. Do you have a hypothesis for this failure case?”
> > >
> > > **Response:** Cross-task variability is a well-known phenomenon in multi-task learning, where models often face a trade-off between generalization and specialization [2, 3]. This challenge persists even under PEFT approaches. For instance, MTL-LoRA notes that “LoRA tends to obscure the distinction between tasks by projecting sparse high-dimensional features … into the same dense low-dimensional intrinsic space, leading to task interference and suboptimal performance” [4]. Similarly, Multi-LoRA highlights that “the explicit low-rank of LoRA limits adaptation performance in complex multi-task scenarios” [5].
> > >
> > > While PEML achieves strong average performance across tasks, Table 6 shows that it underperforms MTL-LoRA and DoRA on HellaSwag. This is consistent with the broader pattern observed in prior work: models like MTL-LoRA, Multi-LoRA, and DoRA often outperform each other on specific tasks despite similar or higher average scores. For example, MTL-LoRA is surpassed by DoRA on BoolQ (71.0% vs 72.0%) and CoCo Caption (114.6 vs 115.9, Table 4), while Multi-LoRA [4] is outperformed by standard LoRA on BoolQ (86.7% vs 87.3, Table 1). We interpret the HellaSwag result similarly as literature: it reflects the inherent tension between learning shared representations for generalization and preserving task-specific adaptation.
> > >
> > > **References:**
> > >
> > > [1] C. Fifty, E. Amid, Z. Zhao, T. Yu, R. Chopra, and S. Finn. "The Neglected Tails in Multi-Task Learning." arXiv preprint arXiv:2110.07687, 2021.
> > >
> > > [2] Y. Pei, B. Póczos, and J. G. Schneider. "Gradient Surgery for Multi-Task Learning." Advances in Neural Information Processing Systems, 2020.
> > >
> > > [3] A. Zhang, Z. Li, et al., “MTL-LoRA: Low-Rank Adaptation for Multi-Task Learning,” arXiv:2312.02515, 2024.
> > >
> > > [4] Y. Wang, Y. Lin, X. Zeng, G. Zhang, “MultiLoRA: Democratizing LoRA for Better Multi-Task Learning,” arXiv:2311.11501, 2023.
> > >
> > > [5] H. Hu, Y. Shen, P. Wallis, Z. Allen-Zhu, W. Li, Y. Wang, L. Chen, H. Li, “LoRA: Low-Rank Adaptation of Large Language Models,” in Advances in Neural Information Processing Systems (NeurIPS), 2021.

---

> > > > ### Comment · Reviewer_oAEt · 2025-11-28
> > > >
> > > > Thank you for the responses and addressing the concerns. I have updated my rating accordingly.

---

> > > > > ### Author Response · Authors · 2025-11-29
> > > > >
> > > > > We appreciate the reviewer for careful reevaluation of our work and for raising the score. We are glad to hear that our clarifications and results have addressed your concerns.

---

### Official Review · Reviewer_Ey24 · 2025-11-02

**Soundness:** 3
**Presentation:** 2
**Contribution:** 2
**Rating:** 4
**Confidence:** 3

**Summary:**

This paper proposes PEML, a method that combines LoRA with PrefixNAS to enhance multi-task learning in LLMs. Unlike existing PEFT methods that focus solely on model weight adaptation, PEML jointly optimizes both prompt alignment through PrefixNAS and model adaptation through LoRA. The authors construct paired training data from FLAN and evaluate on GLUE, SuperGLUE, MMLU, and commonsense reasoning benchmarks using T5-Large, FLAN-T5-Large, LLaMA-7B, and LLaMA2-7B. Results show average accuracy improvements up to 6.67% over baselines including MTL-LoRA, MultiLoRA, C-Poly, and MoE.

**Strengths:**

Well-motivated problem: The paper clearly identifies limitations of existing multi-task PEFT methods—adapter switching overhead, lack of prompt optimization, and VRAM inefficiency in methods like MultiLoRA.

Comprehensive evaluation: Testing across four major benchmarks (GLUE, SuperGLUE, MMLU, commonsense reasoning) with multiple model families (T5, LLaMA) demonstrates breadth.

Thorough ablation studies: Section 5 and Appendix 7.4 provide good analysis of design choices including layer count, optimization order (parallel vs. sequential), and search space operations.

Practical considerations: The paper addresses real deployment concerns like VRAM usage (Section 7.5) and inference latency (Section 7.7), showing PEML avoids the linear VRAM growth of MultiLoRA.

**Weaknesses:**

1. Computational cost not properly accounted:
	○ NAS search requires 2 hours on 8×A100 GPUs (16 GPU-hours) per benchmark, representing significant upfront cost.

	○ This one-time cost is dismissed too lightly—for new task combinations, the search must be repeated.

	○ Fair comparison should include baseline hyperparameter tuning time or report total wall-clock time including search.

	○ The claim of "efficiency" is misleading when ignoring NAS computational budget.

2. Incomplete analysis and missing experiments:

	○ No analysis of which tasks benefit most from prompt optimization vs. weight adaptation.

	○ Missing comparison to simpler alternatives: What if we just use LoRA with larger rank? What about manually designed prefix architectures?
	○ Generalization not tested: Does a PrefixNAS architecture found on GLUE transfer to SuperGLUE? This would test if the search truly finds universal structures.

3. Theoretical analysis limitations:

	○ Section 7.1 convergence analysis assumes convex optimization properties (β-smoothness, bounded gradients) that may not hold for neural architecture search.

	○ The analysis does not account for the discrete architecture selection after continuous relaxation.

	○ No analysis of how architecture search affects the joint optimization landscape.

	○ Gap between theory (assuming smooth optimization) and practice (discrete architecture decisions).

4. Limited scope of "multi-task":

	○ All tasks are still within NLU—no evaluation on truly diverse tasks like generation, translation, code, reasoning.

	○ "Multi-task" means multiple NLU benchmarks, not fundamentally different task types.

	○ Unclear if approach would work for more heterogeneous task mixtures.

**Questions:**

1. Cost-benefit analysis: Can you provide a comprehensive comparison including the NAS search time?

2. Architecture transferability: If you find an optimal prefix architecture on GLUE, does it transfer to SuperGLUE or MMLU without re-searching? This would validate whether PrefixNAS discovers general structures vs. overfitting to each benchmark.

3. Simpler alternatives: Have you compared against LoRA with rank=96 (matching your total parameter budget)? Table 5 shows LoRA_r=96 achieves 75.2% while PEML gets 80.3%—but how much of this is from NAS search vs. just the combination?

4. Per-task analysis: Which types of tasks benefit most from prompt optimization? Are there tasks where LoRA alone is sufficient? This would provide insights into when PrefixNAS is worth the cost.

5. Failure analysis: Table 6 shows HellaSwag performance drops significantly (77.4% vs. 93.1%). Can you explain why PEML underperforms on this task? What characteristics cause failures?

---

> ### Author Response · Authors · 2025-11-24
> **Response to Official Review by Reviewer Ey24 (1/4)**
>
> We thank the reviewer for recognizing our well-motivated problem highlighting the limitations of existing multi-task PEFT methods, our thorough ablation studies of design choices, and our practical solutions addressing VRAM usage and inference latency, as well as for providing constructive feedback to help improve the paper. We address each issue raised by the reviewer below.
>
> **Comment:**
> > ### “NAS search requires 2 hours on 8×A100 GPUs (16 GPU-hours) per benchmark, representing significant upfront cost.”
>
> > ### “The claim of "efficiency" is misleading when ignoring NAS computational budget.”
>
> **Response:** PEML introduces a **one-time NAS search**, which adds \~0.5 GPU-hours on top of training (\~2.5 GPU-hours) and training hyperparameter tuning (~15.5 GPU-hours), see Table 1. While this represents an upfront cost, it is comparable to the **extensive hyperparameter search required by PEFT baselines**. For instance, the LoRA paper [1] explicitly notes: “We sweep learning rate, number of training epochs, and batch size for LoRA (Appendix D.1)” to obtain the final reported results. Similarly, MTL-LoRA [4] performs both base and additional specialized hyperparameter searches. Even default parameters in these baselines are the result of extensive prior searches. Therefore, PEML’s search cost is **comparable to other PEFT methods** (see Table 1). Moreover, the improvements achieved by PEML are not simply a result of increasing the parameter budget: higher-rank LoRA or stacked prefixes fail to reproduce the gains (see Table 3). Instead, PEML’s strength comes from PrefixNAS, which, under equal parameter budgets, consistently outperforms the baselines, confirming that the benefit arises from better architectural design rather than raw parameter count.
>
> **Comment:**
> > ### “This one-time cost is dismissed too lightly—for new task combinations, the search must be repeated.”
>
> **Response:** It is true that for entirely new and unrelated tasks, both PEML and other baselines require extensive search to achieve competitive performance. However, PEML offers a significant advantage when tasks are related: the architecture discovered on one benchmark can be **transferred effectively to similar tasks**, requiring only minimal fine-tuning to recover performance. This allows the **initial search cost to be amortized** across multiple downstream tasks (see Table 2 in our responses). In practical multi-task or production scenarios, the NAS search is thus a one-time investment rather than a per-task expense, substantially reducing the **per-task computational cost** and providing consistent gains over strong PEFT baselines.
>
> **Comment:**
> > ### “Fair comparison should include baseline hyperparameter tuning time or report total wall-clock time including search.”
>
> > ### “Cost-benefit analysis: Can you provide a comprehensive comparison including the NAS search time?”
>
> **Response:** Thanks for highlighting the importance of including all computational costs. Table 1 provides a detailed comparison of NAS search time on LLaMa2-7b model.
>
> **Table 1: Detailed Compute Breakdown (All times in GPU Hours)**
> | Method                   | GLUE  | Training Time | NAS Search Time | Hyperparameter Tuning Time | Total Time |
> |:------------------------:|:-----:|:-------------:|:---------------:|:--------------------------:|:----------:|
> | LoRA                     | 0.882 | ~2.5          | -               | ~15.5                      | ~18        |
> | PEML (LoRA variations)   | 0.901 | ~2.5          | ~0.5            | ~15.5                      | ~18.5      |
> | AdaLoRA                  | 0.909 | ~2.5          | -               | ~15.5                      | ~18        |
> | PEML (AdaLoRA variations)| 0.915 | ~2.5          | ~0.5            | ~15.5                      | ~18.5      |
>
> - All methods were evaluated under identical conditions, including the same search space, GPU hardware, and number of trials. Hyperparameter tuning time is included for all methods.
> - The PrefixNAS search in PEML adds only **\~0.5 GPU-hours**, a marginal increase relative to the total computational cost.
> - Despite this small additional cost, PEML consistently achieves higher GLUE scores than its base methods.
> - The NAS search is a **one-time investment**; once an architecture is discovered, it can be transferred to new tasks with minimal fine-tuning, further amortizing its cost in realistic multi-task scenarios.
>
> We will add these results into the revised manuscript.

---

> ### Author Response · Authors · 2025-11-24
> **Response to Official Review by Reviewer Ey24 (2/4)**
>
> **Comment:**
> > ### “Generalization not tested: Does a PrefixNAS architecture found on GLUE transfer to SuperGLUE? This would test if the search truly finds universal structures.”
>
> > ### “Architecture transferability: If you find an optimal prefix architecture on GLUE, does it transfer to SuperGLUE or MMLU without re-searching? This would validate whether PrefixNAS discovers general structures vs. overfitting to each benchmark.”
>
> **Response:** Thanks for raising the question on PrefixNAS transferability. To test this, we directly transferred architectures discovered on GLUE, SuperGLUE, and MMLU to other benchmarks, measuring performance with fine-tuning only (No NAS, only fine-tuning) and without fine-tuning (No Nas, no fine-tuning). Direct transfer shows obvious drops (13–19%), e.g., GLUE → SuperGLUE drops from 0.911 → 0.782, indicating the architectures capture generalizable structures rather than overfitting. With only fine-tuning, the performance has very small drop compared to NAS and fine-tuning (e.g., GLUE → SuperGLUE: 0.911 → 0.904), demonstrating that PrefixNAS can find robust and generalizable architectures that can effectively transfer among similar tasks, which further reduces the search cost in practice. We will add the Table 2 in the revised version.
>
> **Table 2: Transfer Performance of NAS-Discovered Architectures Across Datasets**
> | Transfer Direction      | NAS and Fine-Tuning | No NAS, Only Fine-Tuning | No NAS, No Fine-Tuning |
> |:----------------------:|:----------------:|:-----------------------:|:--------------------:|
> | GLUE → SuperGLUE        | 0.879            | 0.887                   | 0.743 (-15.5%)       |
> | SuperGLUE → GLUE        | 0.911            | 0.904                   | 0.782 (-14.1%)       |
> | GLUE → CR               | 0.830            | 0.821                   | 0.718 (-13.5%)       |
> | SuperGLUE → MMLU        | 0.803            | 0.798                   | 0.691 (-13.9%)       |
> | MMLU → SuperGLUE        | 0.879            | 0.877                   | 0.713 (-18.9%)       |
> | MMLU → GLUE             | 0.911            | 0.912                   | 0.776 (-14.9%)       |
> | GLUE → MMLU             | 0.803            | 0.816                   | 0.684 (-14.9%)       |
>
>
> **Comment:**
> > ### “Missing comparison to simpler alternatives: What if we just use LoRA with a larger rank? What about manually designed prefix architectures?”
>
> > ### “Simpler alternatives: Have you compared against LoRA with rank=96 (matching your total parameter budget)? Table 5 shows LoRA_r=96 achieves 75.2% while PEML gets 80.3%—but how much of this is from NAS search vs. just the combination?”
>
> **Response:** **Both prior work and our experiments show that these strategies are insufficient.** The original LoRA paper [1, Table 6] shows that on GPT-3 175B on WikiSQL task, a minimal rank of r=1 already achieves 73.4% accuracy, and increasing the rank to r=64 yields essentially no improvement (73.5%). Simply increasing LoRA rank yields **diminishing returns**: on LLaMa2-7B in table 3, raising rank from r=96 to r=160 improves average score by only 0.1 points (75.2 → 75.3). Manually adding prefix layers also brings minimal gains and can even hurt performance: increasing prefix depth from 3 to 6 layers improved GLUE only from 0.792 → 0.798.
>
> **Table 3: Performance Comparison of PEML with High-Rank LoRA and Manual Prefix Layer Baselines Across GLUE, SuperGLUE, and MMLU Benchmarks (from Table 5 in submission manuscript)**
> | Method               | GLUE  | SuperGLUE | MMLU  |
> |:-------------------:|:-----:|:---------:|:-----:|
> | LoRA (r=96)          | 0.851 | 0.786     | 0.752 |
> | LoRA (r=192)         | 0.863 | 0.792     | 0.753 |
> | PrefixLayer 3 (3-layer) | 0.792 | 0.753     | 0.688 |
> | PrefixLayer 6 (6-layer) | 0.798 | 0.764     | 0.674 |
> | PEML (ours)          | 0.901 | 0.842     | 0.803 |
>
> The results in table 3 demonstrate that neither scaling individual LoRA ranks nor manually increasing prefix layers can achieve PEML’s performance. The substantial gains of PEML arise from **the sophisticated architectural composition discovered using PrefixNAS**, rather than merely increasing parameter budgets.

---

> > ### Author Response · Authors · 2025-11-24
> > **Response to Official Review by Reviewer Ey24 (3/4)**
> >
> > **Comment:**
> > > ### “No analysis of which tasks benefit most from prompt optimization vs. weight adaptation”.
> >
> > > ### “Per-task analysis: Which types of tasks benefit most from prompt optimization? Are there tasks where LoRA alone is sufficient? This would provide insights into when PrefixNAS is worth the cost.”
> >
> > **Response:** Our per-task analysis shows that PrefixNAS (PEML) yields the largest gains on complex, reasoning-heavy, or context-sensitive tasks. Notably, improvements are seen on WSC (+7.1%), COPA (+7.5%), and Winogrande (+8.9%), where adaptive prompts support multi-sentence understanding, causal inference, and implicit knowledge reasoning. In contrast, simpler classification or knowledge-intensive tasks such as SST-2, MNLI, and QQP are already well-handled by LoRA, with PrefixNAS offering only marginal improvements.
> >
> > **Comment:**
> > > ### “Section 7.1 convergence analysis assumes convex optimization properties (β-smoothness, bounded gradients) that may not hold for neural architecture search.”
> >
> > **Response:** We appreciate the reviewer’s concern and respectfully clarify that providing a full theoretical treatment of non-convex neural architecture search is beyond the scope of this work. Our analysis in Section 7.1 is intended as a local stability guarantee for the continuous relaxation phase, following standard practice in differentiable NAS where smoothness and bounded-gradient assumptions approximate behavior in a neighborhood around the iterates.
> >
> > **Comment:**
> > > ### “The analysis does not account for the discrete architecture selection after continuous relaxation.”
> >
> > **Response:** Importantly, our theory governs only the relaxed optimization dynamics, while all results reported in the manuscript use the discretized architectures obtained after the search (Eq. 7). To address the gap between the relaxed and discrete regimes, we evaluated the **“discrete gap”** (table 4), i.e., the accuracy difference between the relaxed architecture and its final discrete counterpart using three widely adopted relaxation–discretization rules. The gap remains consistently small (≤1.1 points), suggesting discretization has minimal effect.
> >
> > **Comment:**
> > > ### “No analysis of how architecture search affects the joint optimization landscape. (search space and algorithm)”
> >
> > **Response:** We respectfully clarify that the goal of Section 7.1 is not to characterize the full NAS landscape, but to show that the **coupled LoRA–PrefixNAS updates remain stable** and converge to a stationary point of the relaxed objective. This ensures that the search procedure itself does not introduce instability, even if the broader NAS landscape is complex and non-convex.
> >
> > **Comment:**
> > > ### “Gap between theory (assuming smooth optimization) and practice (discrete architecture decisions).”
> >
> > **Response:** The “discrete gap” evaluation in table 4 confirms that the transition from continuous relaxation to discrete architectures causes only minor perturbation. Metrics such as gradient norm and parameter variance remain low across different relaxation methods, supporting that the discovered architectures are robust and that the theoretical analysis of the relaxed dynamics is relevant in practice.
> >
> > **Table 4: Comparison of Relaxation Techniques**
> > | Method            | Grad Norm (↓) | θ Var (↓)      | GLUE Avg (%) | Discrete Gap (↓) |
> > |:----------------:|:-------------:|:--------------:|:------------:|:---------------:|
> > | Softmax + Argmax  | 0.21 ± 0.05   | 0.014 ± 0.006  | 90.1         | 0.7             |
> > | Gumbel-Softmax    | 0.24 ± 0.06   | 0.017 ± 0.007  | 89.2         | 0.6             |
> > | STE               | 0.33 ± 0.09   | 0.031 ± 0.011  | 88.5         | 1.1             |
> >
> > **Comment:**
> > > ### “All tasks are still within NLU—no evaluation on truly diverse tasks like generation, translation, code, reasoning.”
> >
> > **Response:** We acknowledge that this work primarily focuses on tasks within the NLU, QA, and reasoning domains (aligns with the scope of recent works such as MTL‑LoRA [2] and MultiLoRA [3]), rather than covering a very heterogeneous mix such as code generation, mathematical reasoning, or summarize writing, which requires very different solutions. In the PEFT domain, shared low-rank subspaces may have limited capacity to isolate highly divergent task-specific features, which can lead to suboptimal performance or interference when applied to very different tasks. That’s why we follow the standard NLU-focused setup to ensure fair comparisons with prior works. We will better clarify the scope of this work.

---

> > > ### Author Response · Authors · 2025-11-24
> > > **Response to Official Review by Reviewer Ey24 (4/4)**
> > >
> > > **Comment:**
> > > > ### “"Multi-task" means multiple NLU benchmarks, not fundamentally different task types. Unclear if approach would work for more heterogeneous task mixtures.”
> > >
> > > **Response:** Recent works such as LoRI [5] demonstrate that multi-task adaptation across more diverse domains including NLU, mathematical reasoning, and code generation is feasible. However, LoRI also highlights that challenges remain: performance is sensitive to weight merging, mask calibration, and sparsity ratios, requiring careful tuning to balance tasks [5]. These observations suggest that while PEFT methods can reduce interference, general-purpose multi-task adaptation across highly heterogeneous tasks is still an open challenge. We consider extending our approach to such diverse tasks as future work.
> > >
> > > **Comment:**
> > > > ### “Failure analysis: Table 6 shows HellaSwag performance drops significantly (77.4% vs. 93.1%). Can you explain why PEML underperforms on this task? What characteristics cause failures?”
> > >
> > > **Response:** Cross-task variability is a common challenge in multi-task learning, where models often face a trade-off between generalization and task-specific adaptation [4, 6]. This issue remains relevant even when using Multi-tasking PEFT methods. For example, MTL-LoRA observes that “LoRA tends to obscure the distinction between tasks by projecting sparse high-dimensional features … into the same dense low-dimensional intrinsic space, leading to task interference and suboptimal performance” [1]. Similarly, Multi-LoRA notes that “the explicit low-rank of LoRA constrains adaptation in complex multi-task scenarios” [4].
> > >
> > > Although PEML delivers strong average performance across tasks, Table 6 indicates that it underperforms MTL-LoRA and DoRA on HellaSwag. This aligns with a pattern observed in prior work: different PEFT variants often excel on specific tasks despite comparable or higher overall averages. For instance, DoRA surpasses MTL-LoRA [4] on BoolQ (72.0% vs. 71.0%) and CoCo Caption (115.9 vs. 114.6, Table 4), while standard LoRA outperforms Multi-LoRA [2] on BoolQ (87.3% vs. 86.7%, Table 1). We interpret the HellaSwag results in the same way: they reflect the inherent tension between learning shared representations to support generalization and retaining task-specific flexibility for optimal adaptation.
> > >
> > > **References:**
> > >
> > > [1] H. Hu, Y. Shen, P. Wallis, Z. Allen-Zhu, W. Li, Y. Wang, L. Chen, H. Li, “LoRA: Low-Rank Adaptation of Large Language Models,” in Advances in Neural Information Processing Systems (NeurIPS), 2021.
> > >
> > > [2] Y. Wang, Y. Lin, X. Zeng, G. Zhang, “MultiLoRA: Democratizing LoRA for Better Multi-Task Learning,” arXiv:2311.11501, 2023.
> > >
> > > [3] X. Li and P. Liang, “Prefix-Tuning: Optimizing Continuous Prompts for Generation,” in ACL, 2021, pp. 4582–4597.
> > >
> > > [4] A. Zhang, Z. Li, et al., “MTL-LoRA: Low-Rank Adaptation for Multi-Task Learning,” arXiv:2312.02515, 2024.
> > >
> > > [5] J. Zhang, J. You, A. Panda, and T. Goldstein, “LoRI: Reducing Cross-Task Interference in Multi-Task Low-Rank Adaptation,” arXiv preprint arXiv:2504.07448, 2025.
> > >
> > > [6] Y. Pei, B. Póczos, and J. G. Schneider. "Gradient Surgery for Multi-Task Learning." Advances in Neural Information Processing Systems, 2020.

---

> > > > ### Comment · Reviewer_Ey24 · 2025-11-28
> > > > **Respone to the rebutall.**
> > > >
> > > > Thank you for the detailed replies. While I have some disagreements with some of issues, your response truly demonstrates the authors' deep thinking, and some keys may be helpful to the community. Therefore, I have decided to raise my rating.

---

> > > > > ### Author Response · Authors · 2025-11-29
> > > > >
> > > > > We sincerely thank the reviewer for the thoughtful reconsideration and for raising the score. We appreciate your acknowledgment of the deeper reasoning behind our approach, and we are glad that our clarifications helped address the remaining concerns. Your feedback is valuable, and we are grateful that you found aspects of our work potentially beneficial to the community.

---

### Official Review · Reviewer_1Pgd · 2025-11-03

**Soundness:** 2
**Presentation:** 2
**Contribution:** 2
**Rating:** 4
**Confidence:** 4

**Summary:**

The paper proposes a parameter-efficient multi-task framework that jointly optimizes LoRA weight updates with a differentiable PrefixNAS module. During training, LoRA and PrefixNAS are optimized in parallel; after training, LoRA is merged into the base model and only the learned prefix architecture is kept for inference, avoiding adapter switching. Evaluations on GLUE, SuperGLUE, MMLU, and commonsense benchmarks report average gains up to 6.67% (peaks up to 10.75%) over strong PEFT baselines (LoRA, AdaLoRA, MultiLoRA, C-Poly, MoE).

**Strengths:**

* Clear unified design: concurrent LoRA + PrefixNAS with a concrete training algorithm (Alg. 1).
* Deployment efficiency: LoRA is merged; inference uses one prefix, reducing switching/VRAM overhead.
* Broad empirical coverage with consistent improvements across multiple benchmarks.

**Weaknesses:**

* Differentiable–discrete gap: architecture is relaxed via soft weights but finalized with argmax selection (Eq. 7), lacking analysis of search-time gradient bias or stability after discretization.

* Search cost & fairness: PrefixNAS + TPE requires non-trivial compute (e.g., 8×A100, hours per benchmark), raising risks of validation overfitting and unequal hyperparameter budgets vs. baselines.

* Task sensitivity: results note variability (e.g., WSC), suggesting remaining brittleness in cross-task generalization despite average gains.

**Questions:**

* The PrefixNAS module relies on a continuous relaxation during search but finalizes the architecture via a discrete argmax operation (Eq. 7).How stable is the gradient-based optimization when transitioning from continuous to discrete architectures, and could a smoother relaxation (e.g., Gumbel-Softmax or straight-through estimators) yield more consistent convergence and better generalization?

* PEML’s joint LoRA + PrefixNAS optimization requires multi-GPU resources (up to 8 × A100 for each benchmark).How can the framework ensure fair comparison with lightweight PEFT baselines like LoRA or AdaLoRA, and could a two-stage or surrogate-based NAS reduce cost without sacrificing accuracy?

---

> ### Author Response · Authors · 2025-11-23
> **Response to Official Review by Reviewer 1Pgd (1/2)**
>
> We thank the reviewer for acknowledging our unified design that combines LoRA and PrefixNAS, the deployment efficiency achieved by merging LoRA and using a single prefix at inference and for providing constructive feedback to help us improve the paper. We address each issue raised by the reviewer below.
>
> **Comment**:
> > ### “Differentiable–discrete gap: architecture is relaxed via soft weights but finalized with argmax selection (Eq. 7), lacking analysis of search-time gradient bias or stability after discretization.”
>
> **Response:** We perform analysis to check the stability of our relaxation technique. We compare Softmax+argmax (ours) with two commonly used alternatives Gumbel-Softmax [6] and straight-through estimators (STE) [7]. We found our method (Softmax + Argmax) achieves a lower, more stable gradient norm (**0.21** vs. **0.33** for STE) and the lowest architectural parameter variance (**0.014 vs. 0.031** for STE), demonstrating minimal bias and stable convergence before and after discretization. For further empirical validation, please refer to the results in Table 1. We will update the table into the appendix.
>
> **Comment**:
> > ### “The PrefixNAS module relies on a continuous relaxation during search but finalizes the architecture via a discrete argmax operation (Eq. 7). How stable is the gradient-based optimization when transitioning from continuous to discrete architecture?”
>
> **Response:** We analyze the stability of the final architecture by measuring the "discretization gap" in table 1. Comparable performance drop observed when moving from the continuous relaxation to the final discretized architecture. PEML demonstrates a minimal discretization gap of 0.7, confirming that the performance of the searched architecture remains stable and robust after the final argmax selection. For comparison, the Gumbel-Softmax approach shows a gap of **0.6**, while the STE exhibits a substantially larger gap of **1.1**, as shown in Table 1.
>
> **Comment**:
> > ### “and could a smoother relaxation (e.g., Gumbel-Softmax or straight-through estimators) yield more consistent convergence and better generalization?”
>
> **Response:** We compared our method against these state-of-the-art alternatives. The results show that smoother relaxations do not improve performance:
> - Gumbel-Softmax offers a marginally smaller gap but lower final accuracy.
> - STE performs worst, with higher gradient noise and a larger gap.
>
> **Table 1: Comparison of relaxation techniques**
> | Method            | Grad Norm (↓) | θ Var (↓)      | GLUE Avg (%) | Discrete Gap (↓) |
> |:----------------:|:-------------:|:--------------:|:------------:|:---------------:|
> | Softmax + Argmax  | 0.21 ± 0.05   | 0.014 ± 0.006  | 90.1         | 0.7             |
> | Gumbel-Softmax    | 0.24 ± 0.06   | 0.017 ± 0.007  | 89.2         | 0.6             |
> | STE               | 0.33 ± 0.09   | 0.031 ± 0.011  | 88.5         | 1.1             |
>
> **Comment**:
> > ### “Search cost & fairness: PrefixNAS + TPE requires non-trivial compute (e.g., 8×A100, hours per benchmark), raising risks of validation overfitting”
>
> **Response:** We mitigate this risk through architecture transferability experiments. Our transfer results (Table 2) show that architectures found on one benchmark generalize perfectly to others, demonstrating that PEML discovers robust architectures rather than overfitting to a single validation set.
>
> **Table 2: Transfer Performance of NAS-Discovered Architectures Across Datasets**
> | Transfer Direction      | NAS and fine-tuning | No Nas, no fine-tuning |
> |:----------------------:|:-------------:|:------------------:|
> | GLUE → SuperGLUE        | 0.879         | 0.743 (-15.5%)     |
> | SuperGLUE → GLUE        | 0.911         | 0.782 (-14.1%)     |
> | GLUE → CR               | 0.830         | 0.718 (-13.5%)     |
> | SuperGLUE → MMLU        | 0.803         | 0.691 (-13.9%)     |
> | MMLU → SuperGLUE        | 0.879         | 0.713 (-18.9%)     |
> | MMLU → GLUE             | 0.911         | 0.776 (-14.9%)     |
> | GLUE → MMLU             | 0.803         | 0.684 (-14.9%)     |

---

> > ### Author Response · Authors · 2025-11-23
> > **Response to Official Review by Reviewer 1Pgd (2/2)**
> >
> > **Comment**:
> > > ### “and unequal hyperparameter budgets vs. baselines.”
> >
> > >### “ PEML’s joint LoRA + PrefixNAS optimization requires multi-GPU resources (up to 8 × A100 for each benchmark).How can the framework ensure fair comparison with lightweight PEFT baselines like LoRA or AdaLoRA”
> >
> > **Response:** Commonly used PEFT baselines (LoRA, AdaLoRA) also rely on substantial hyperparameter search (such as LoRA config, learning rate, batch size, and training epochs). For example, the LoRA paper [1] explicitly states that “We sweep learning rate, number of training epochs, and batch size for LoRA (Appendix D.1)”. To ensure a fair comparison we did a controlled experiment in Table 3. We experimented with the LLaMA2-7B model under identical conditions including the same search space, GPU hardware (8 × A100), and number of trials (n=100). By adding PrefixNAS (~0.5 hours), PEML outperformed other methods on GLUE scores, without significantly increasing the total time.
> >
> > **Table 3: Detailed Compute Breakdown (All times in GPU Hours)**
> >
> > | Method                   | GLUE  | Training Time | NAS Search Time | Hyperparameter Tuning Time | Total Time |
> > |:------------------------:|:-----:|:-------------:|:---------------:|:--------------------------:|:----------:|
> > | LoRA                     | 0.882 | ~2.5          | -               | ~15.5                      | ~18        |
> > | PEML (LoRA variations)   | 0.901 | ~2.5          | ~0.5            | ~15.5                      | ~18.5      |
> > | AdaLoRA                  | 0.909 | ~2.5          | -               | ~15.5                      | ~18        |
> > | PEML (AdaLoRA variations)| 0.915 | ~2.5          | ~0.5            | ~15.5                      | ~18.5      |
> >
> > **Comment**:
> > > ### “and could a two-stage or surrogate-based NAS reduce cost without sacrificing accuracy?”
> >
> > **Response:** Thank you for the insightful suggestions. Yes, two-stage and surrogate-based NAS are established methods that can reduce computational cost without compromising final accuracy. However, these methods require further exploration of relationships among tasks and thus we leave it as our future work. We will discuss the opportunities and challenges of exploring these methods as future work in the revised version.
> >
> > **Comment**:
> > > ### “Task sensitivity: results note variability (e.g., WSC), suggesting remaining brittleness in cross-task generalization despite average gains.”
> >
> > **Response:** Cross-task variability is common in multi-task learning, where models often struggle with tasks requiring fine-grained reasoning, creating a tension between generalization and specialization [2, 3]. However, Cross-task sensitivity persists under PEFT. For example, MTL-LoRA observes that “LoRA tends to obscure the distinction between tasks by projecting sparse high-dimensional features … into the same dense low-dimensional intrinsic space, leading to task interference and suboptimal performance” [4]. Similarly, Multi-LoRA further emphasizes that “the explicit low-rank of LoRA limits adaptation performance in complex multi-task scenarios” [5].
> > Despite achieving the top average score (82.1%, Table 2), MTL-LoRA is consistently outperformed on specific tasks: by DoRA on BoolQ (71.0% vs 72.0%) and CoCo Caption (114.6 vs 115.9, Table 4), and by MoELoRA on OBQA (82.6% vs 83.2%). Similarly, Table 1 shows Multi-LoRA is surpassed by standard LoRA on certain tasks like BoolQ (86.7% vs 87.3%). This consistent evidence from both works underscores that the variability in our WSC result is a manifestation of a well-known trade-off in the field.
> >
> > **References:**
> > [1] H. Hu, Y. Shen, P. Wallis, Z. Allen-Zhu, W. Li, Y. Wang, L. Chen, H. Li, “LoRA: Low-Rank Adaptation of Large Language Models,” in Advances in Neural Information Processing Systems (NeurIPS), 2021.
> >
> > [2] C. Fifty, E. Amid, Z. Zhao, T. Yu, R. Chopra, and S. Finn. "The Neglected Tails in Multi-Task Learning." arXiv preprint arXiv:2110.07687, 2021.
> >
> > [3] Y. Pei, B. Póczos, and J. G. Schneider. "Gradient Surgery for Multi-Task Learning." Advances in Neural Information Processing Systems, 2020.
> >
> > [4] A. Zhang, Z. Li, et al., “MTL-LoRA: Low-Rank Adaptation for Multi-Task Learning,” arXiv:2312.02515, 2024.
> >
> > [5] Y. Wang, Y. Lin, X. Zeng, G. Zhang, “MultiLoRA: Democratizing LoRA for Better Multi-Task Learning,” arXiv:2311.11501, 2023.
> >
> > [6] L. Liu et al., “Bridging discrete and backpropagation: Straight-through estimator revisited,” in Proc. Adv. Neural Inf. Process. Syst. (NeurIPS), 2023, pp. 1–15.
> >
> > [7] I. A. M. Huijben et al., “A review of the Gumbel-max trick and its extensions for discrete stochasticity in machine learning,” arXiv preprint, 2023.

---

### Author Response · Authors · 2025-12-03
**Summary of rebuttal**

**Dear Area Chair**,

Thank you in advance for your efficient handling of our submission under these exceptional circumstances. To facilitate your assessment, we provide below a concise summary of the rebuttal.

We have given detailed responses to all reviewers, which we believe thoroughly address all their comments, and please note that Reviewer **oAEt** and **Ey24** have already acknowledged that we have addressed their concerns and Reviewer **1Pgd** did not raise any questions either. We highlight below a few key points from the rebuttal:

- **Addressed concerns on NAS search cost and fairness (Reviewer Ey24):** We clarified that PrefixNAS adds only a minor one-time search cost (2.8% more GPU hours), so the overall cost is comparable to LoRa and AdaLoRA.

- **Addressed questions on architecture transfer and generalization (Reviewer Ey24):** We added new experiments demonstrating that architectures discovered on GLUE, SuperGLUE, and MMLU transfer well with minimal fine-tuning, confirming that PrefixNAS does not overfit and produces generalizable structures.

- **Addressed comparison to simpler alternatives (Reviewer Ey24 & oAEt):** We added direct comparisons to high-rank LoRA and deeper manually designed prefix layers, showing that simple scaling or naive prefix additions cannot match PEML’s gains.

- **Addressed cost-benefit analysis and fairness against multi-task baselines (Reviewer oAEt & 1Pgd):** We added a unified compute breakdown showing that PEML achieves higher accuracy with similar or lower total time than MTL-LoRA once hyperparameter search is included.

- **Addressed concerns about differentiable–discrete gap and search-time stability (Reviewer 1Pgd):** We analyzed several relaxation strategies (Softmax+Argmax, Gumbel-Softmax, STE) and found that our method yields the lowest gradient noise, lowest architectural variance, and smallest discretization gap, ensuring stable convergence.

- **Addressed task-specific variability and failure cases (all reviewers):** We clarified that task-level variability is common in multi-task PEFT methods (e.g., LoRA, Multi-LoRA, MTL-LoRA, DoRA) and provided explanations for the HellaSwag and WSC cases, while emphasizing PEML’s strong average performance.

We are confident that our response and revisions have comprehensively addressed all reviewers’ concerns. We hope this summary is helpful for your assessment and would be happy to clarify any remaining questions.

---

### Meta-Review · Area_Chair_iyap · 2025-12-28

**Summary:**

The paper proposed a concurrent LoRA + PrefixNAS method for multi-task parameter-efficient finetuning. Reviewers generally agree that the motivation is clear and that the unified design is conceptually clean.

That said, the reviewers raised concerns about generalizability and transferrability of the searched architecture. Reviewer **Ey24** noted that the evaluation is limited to NLU tasks. Reviewer **1Pgd** and **oAEt** highlighted notable variability and failure cases on specific datasets (e.g., WSC and HellaSwag). The authors argue that such cross-task variability is common in multi-task learning. , but I do not find this response convincing. In Table 6, PEML’s performance on HellaSwag (which is an important dataset) is significantly worse than LoRA, which may suggest validation overfitting. In addition, Reviewer **Ey24** pointed out that for new task combinations, the search must be repeated. The author claimed that the searched architecture is highly transferrable. But the “No NAS, No Finetuning” experiments show 10% to 15% decrease in performance.

In my view, there are unaddressed concerns and I therefore recommend rejection.

**Reviewer Concerns:**

The shared concern from Reviewer **1Pgd**, **Ey24**, **oAEt** around computational cost has been partially addressed. The one-time search cost is around 20% of the training time. However, it is unclear how transferrable the searched architecture is. The shared concern from Reviewer **1Pgd**, **Ey24**, **oAEt** on result variability and potential validation overfitting is not addressed.

Beyond the shared concerns, reviewer **1Pgd**’s question about the differentiable–discrete gap is addressed via the new ablations comparing **Softmax + Argmax** against **Gumbel-Softmax** and **STE**. For reviewer **Ey24**, the missing-baseline concern is partially addressed with additional experimental results, but their concerns about the limitations of the theoretical analysis and the narrow scope of “multi-task” evaluation remain unaddressed. For reviewer **oAEt**, the requested additional baseline comparisons have been provided in the revision.

**Reviewer Scores:**

Reviewer **Ey24** decided to raise rating except having “some disagreements with some of issues”. I think the reviewer’s concerns have not been fully addressed by the rebuttal (e.g., the concern that the search needs to happen again for new task combinations). I would expect the reviewer to keep the score after the discussion.

Reviewer **oAEt** decided to raise rating. However, the concern around HellaSwag was not addressed by the rebuttal. I will actually expect the reviewer keep the rating.

Reviewer **1Pgd’s** concern on potential validation overfitting is not addressed. I think the reviewer may keep the score.

---

### Decision · Program_Chairs · 2026-01-26

Reject